# Circulating cell-free DNA methylation patterns indicate cellular sources of allograft injury after liver transplant

Megan E. McNamara [1], Sidharth S. Jain[1], Kesha Oza[2,3], Vinona Muralidaran[2], Amber J. Kiliti [1], A. Patrick McDeed[1], Digvijay Patil[2], Yuki Cui[2], Marcel O. Schmidt[1], Anna T. Riegel[1], Alexander Kroemer[2] ✉ & Anton Wellstein [1] ✉

Post-transplant complications reduce allograft and recipient survival. Current approaches for detecting allograft injury non-invasively are limited and do not differentiate between cellular mechanisms. Here, we monitor cellular damages after liver transplants from cell-free DNA (cfDNA) fragments released from dying cells into the circulation. We analyzed 130 blood samples collected from 44 patients at different time points after transplant. Sequence-based methylation of cfDNA fragments were mapped to an atlas of cell-type-specific DNA methylation patterns derived from 476 methylomes of purified cells. For liver cell types, DNA methylation patterns and multi-omic data integration show distinct enrichment in open chromatin and functionally important regulatory regions. We find that multi-tissue cellular damages post-transplant recover in patients without allograft injury during the first post-operative week. However, sustained elevation of hepatocyte and biliary epithelial cfDNA within the first month indicates early-onset allograft injury. Further, cfDNA composition differentiates amongst causes of allograft injury indicating the potential for non-invasive monitoring and intervention.

Liver transplant is the standard-of-care for patients with end-stage liver disease and is the second most common solid-organ transplant after the kidney[1]. Despite improved survival rates, there is still a high prevalence of complications contributing to perioperative mortality post-liver transplant, mostly occurring within the first month[2,3]. Unfortunately, current non-invasive biomarkers have a limited scope and fail to identify cellular causes of allograft injury[4]. Thus, tissue biopsy is still the gold-standard to confirm diagnosis and monitor response to treatment. The analysis of cell-free DNA (cfDNA) in the circulation is an alternative to conventional biomarkers. CfDNA consists of fragments shed by dying cells throughout the body and its analysis can serve as a non-invasive approach for monitoring

allograft as well as host tissue changes at a cellular level following liver transplant[5–11].

Genetic differences between donor and recipient (SNPs) have been used to identify donor-derived cfDNA (dd-cfDNA) originating from the transplanted allograft to serve as a predictive biomarker of allograft injury and rejection[8,12–18]. However, there are many situations when genetic differences cannot be used to identify allograft-derived DNA; for example, when the genotype is unknown, multiple genotypes exist in the host, and when the donor is closely related to the recipient[13]. Instead, epigenetic modifications can be used to identify cfDNA that is recipient or allograft-derived by using tissue- and cell-type specific marks that are independent of genotype differences

[1]Department of Oncology, Lombardi Comprehensive Cancer Center, Georgetown University, Washington, DC, USA. [2]MedStar Georgetown Transplant Institute, MedStar Georgetown University Hospital and Center for Translational Transplant Medicine, Georgetown University Medical Center, Washington, DC, USA. [3]Department of General Surgery, MedStar Georgetown University Hospital, Washington, DC, USA. ✉e-mail: alexander.kroemer@gunet.georgetown.edu; anton.wellstein@georgetown.edu

between the donor and recipient[19–29]. Allograft injury can thus be detected in an organ specific fashion, which is of critical importance in recipients of multi-organ transplants and recipients of hematopoietic cell transplant (HCT) who develop Graft-versus-Host disease (GvHD)[30–32]. Also, the allograft as well as recipient organs are impacted by the transplant process as well as by subsequent treatments that can lead to tissue damage and remodeling. Primary injuries or secondary changes stemming from tissue repair can be quantified from cfDNA to indicate cell-type-specific damage[1–4,33,34].

DNA methylation is highly cell-type-specific and has been found to reveal the origins of tissue damages and altered cell turnover from cfDNA samples in a wide-range of applications[27,28,30,31,35–45]. DNA methylation patterns are stable epigenetic marks for cells maintained throughout DNA replication and cell proliferation[46]. Cell-type-specific DNA methylation established early-on during development has been found to be highly conserved across individuals, irrespective of age and disease status[21,22,46–50]. Also, cell-free DNA methylation can reflect intervention-related changes over time through analyses of serially collected blood samples[38]. Fragment-level deconvolution of methylation sequencing data allows for increased sensitivity and specificity of signal localization using CpG pattern analysis of individual cfDNA molecules[35,38,51–53]. Likewise, hybridization capture to CpG-rich DNA segments maximizes sequencing depth while still maintaining comprehensive coverage[38].

Here, we utilize circulating, cell-free DNA methylation patterns to monitor cellular damages after liver transplant, impacting the allograft tissue as well as the recipient's organs. We expand existing cell-type-specific DNA methylation atlases to include non-parenchymal cell-types from the liver, including hepatic stellate, endothelial, liver-resident immune, and biliary epithelial cells. Then, we perform capture-sequencing of cell-free DNA from serial blood samples and evaluate multi-tissue cellular damages after liver transplant. We find that sustained elevation of hepatocyte and biliary epithelial cfDNA within the first post-operative month is indicative of allograft injury. In addition, we show that there are significant changes in cfDNA composition corresponding to different allograft injury patterns at time of tissue-biopsy-proven diagnosis. Thus, changes in cell-free DNA methylation patterns can non-invasively indicate cellular sources of allograft injury in liver transplant patients.

## Results

### Cellular damages after liver transplant indicated by cell-free DNA methylation patterns in the circulation

To monitor cellular damages after liver transplant, we collected serial serum samples from 28 adult liver transplant patients during the peri-transplant time period and profiled cfDNA methylation from samples at predetermined timepoints up to one month after transplant (n = 100 samples). We also collected samples from patients experiencing complications and added phenotype-matched samples from an additional 16 patients at the time of for-cause liver biopsy (FC-bx) used to identify allograft injury (n = 30 samples; patient characteristics in Supplementary Data 1). Cell-free DNA fragments isolated from these 130 serum samples were bisulfite treated, enriched for sequences of interest by methylome-wide hybridization capture and subjected to sequence analysis (Fig. 1). The tissue and cell type origins of cfDNA fragments in the circulation were mapped to an expanded atlas of cell-type-specific DNA methylation described next to infer tissue damages and differentiate amongst causes of allograft injury.

### Characterization of liver cell-type-specific epigenomes to expand the sequencing-based DNA methylation atlas of healthy tissues

To identify cellular origins of cfDNA fragments in the circulation, we expanded the existing cell-type-specific DNA methylation atlas to

liver cell-types relevant for injury and repair and generated methylome-sequencing data for hepatic stellate, liver endothelial, biliary epithelial, and liver-resident immune cell populations. In addition, we also included published whole genome bisulfite sequencing (WGBS) data from purified healthy human cell-types[35,38]. This resulted in curation of over 450 WGBS datasets encompassing over 40 cell-types from diverse populations of donors (Supplementary Data 2) to generate a reference methylome atlas[35,38]. Briefly, we first segmented the data into homogenously methylated blocks where DNA methylation levels at adjacent CpG sites were highly correlated across different cell types. Then, we restricted the analysis to the 364,268 blocks covered by our hybridization capture panel used in the analysis of cfDNA in human serum (captures 80 Mb, ~20% of CpGs). Average methylation was calculated within blocks of at least three CpG sites and unsupervised clustering analysis was performed for the top 10% variable blocks across all samples. We found that with the additional data incorporated, samples still clustered strongly by cell type and developmental lineage (Supplementary Fig. 1d). Notably, parenchymal and non-parenchymal liver cell methylomes did not cluster together. Instead, samples clustered with other cell-types of the same lineage, independent of the germ layer origin of their tissues of residence. Interestingly, biliary epithelial samples isolated from intrahepatic ducts and the gallbladder (biliary-small-ductal) demonstrated distinct methylation patterns compared to biliary epithelial samples isolated from the larger main hepatic, common bile and pancreatic ducts (biliary-large-ductal) (Supplementary Fig. 1b, c).

Based on the unsupervised clustering analysis, we grouped the reference WGBS data into 20 groups for downstream analysis (Supplementary Data 2). We identified cell-type-specific differentially methylated blocks (DMBs) within these groups taking a one-vs-all approach[38]. The co-methylation status of neighboring CpG sites in each block distinguished amongst all cell types included in the final groups (Supplementary Data 3). The heatmap in Fig. 2a depicts the top 50 blocks with the highest score for each cell-type and the top hepatocyte-specific blocks are emphasized in Fig. 2b, c. The methylation sequencing approach taken here allowed for assessment of fragment-level methylation patterns rather than the limited single-site resolution of methylation arrays (Fig. 2d)[6]. Whereas bulk tissue analyses average the methylation status amongst all cell-types, purified cell-specific methylome analysis allowed for discovery of features critical to the identity of non-parenchymal cell-types that contribute only few cells to the overall population and therefore would otherwise be missed[54,55].

### Liver cell-type-specific hypomethylated blocks coincide with cell-type-specific chromatin accessibility and H3K27ac binding

The cell-type-specific DMBs identified using the expanded WGBS reference data resembled those of previously published methylation atlases, being largely hypomethylated, intragenic and annotated to genes relevant for cell function and identity (Supplementary Figs. 1a, 2f and 3a; Supplementary Data 4)[35,38,41]. Notably, cell-type-specific hypermethylated DMBs were much less frequent (17% on average) and enriched for CpG islands compared to cell-type-specific hypomethylated DMBs that were located in relatively CpG-depleted, GC-low regions characteristic of programmed demethylation occurring at enhancers (Supplementary Fig. 2a)[35,47,56]. Indeed, the majority of liver cell-type-specific hypomethylated DMBs were enhancers by chromHMM annotations (Fig. 3c). In contrast, the majority of liver cell-type-specific hypermethylated DMBs were annotated to bivalent TSS/enhancers and repressive Polycomb targets (Supplementary Fig. 2d). This matches with the function of cell-type-specific hypermethylation in repressing genes associated with embryonic stem cell pluripotency to stabilize cellular differentiation during development (Supplementary Fig. 2e)[57–59]. To further explore the liver cell-type-specific DMBs identified, we generated and compiled

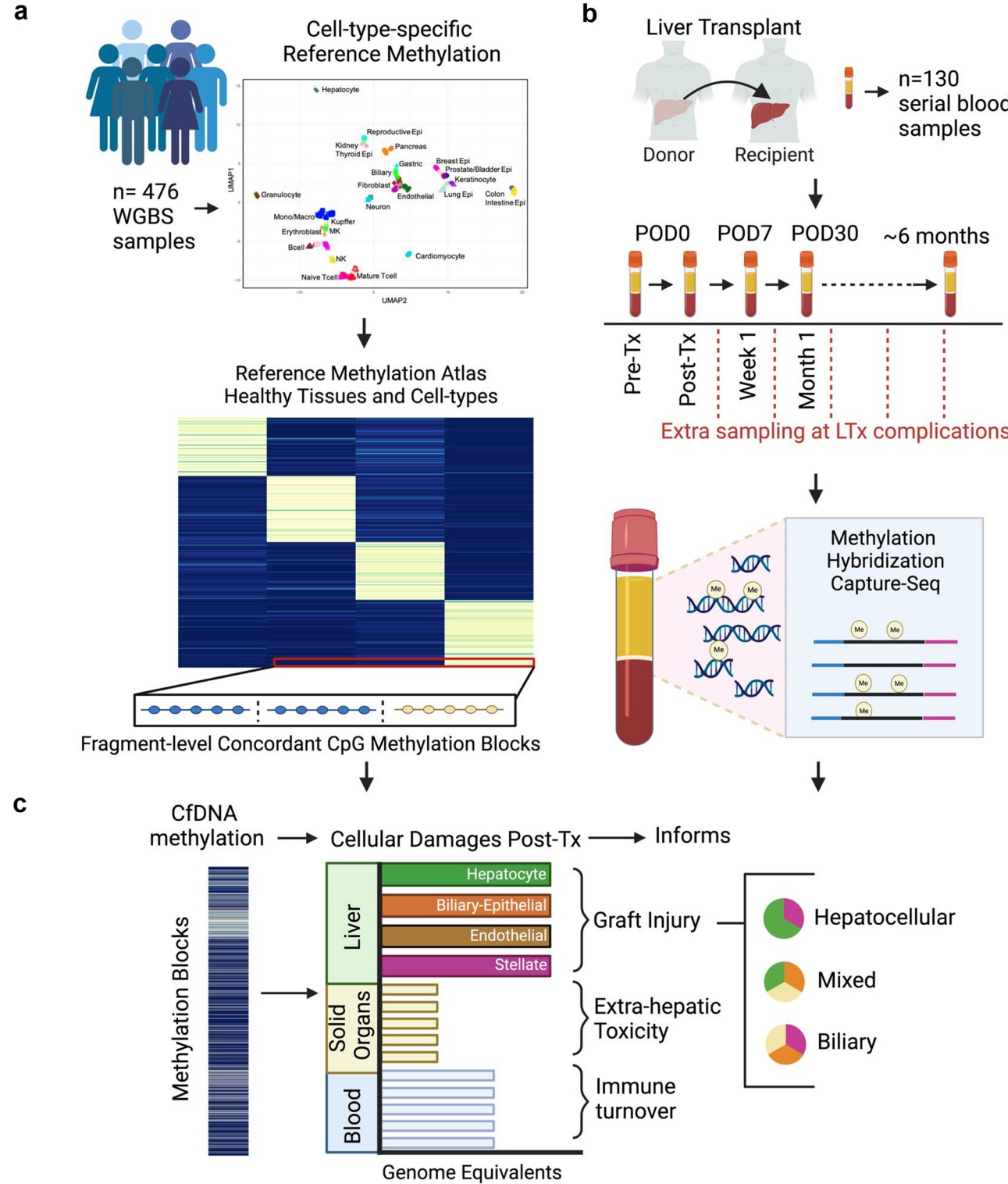

**Fig. 1 | Study overview using cell-free DNA methylation patterns in blood to monitor cellular damages after liver transplant. a** Cell-type-specific DNA methylation patterns were identified from reference data consisting of 476 methylomes of purified cells within healthy tissues. Methylation patterns were validated to show distinct enrichment in open chromatin and functionally important regulatory regions and then used to trace the origins of patient cfDNA fragments. **b** Serial serum samples were collected from 28 patients before and after liver transplant at predetermined time points ($n = 100$ samples) during the first month. We also collected serum samples from an additional 16 patients at the time of liver-biopsy proven allograft injury ($n = 30$ samples). Cell-free DNA (cfDNA) methylome profiling of serum samples was performed using hybridization capture-sequencing of bisulfite-treated cfDNA. **c** Cellular damages of the transplanted organ as well as other recipient organs were quantified to monitor organism-wide impact. Figure created in BioRender (2025) [https://BioRender. com/i5lj954].

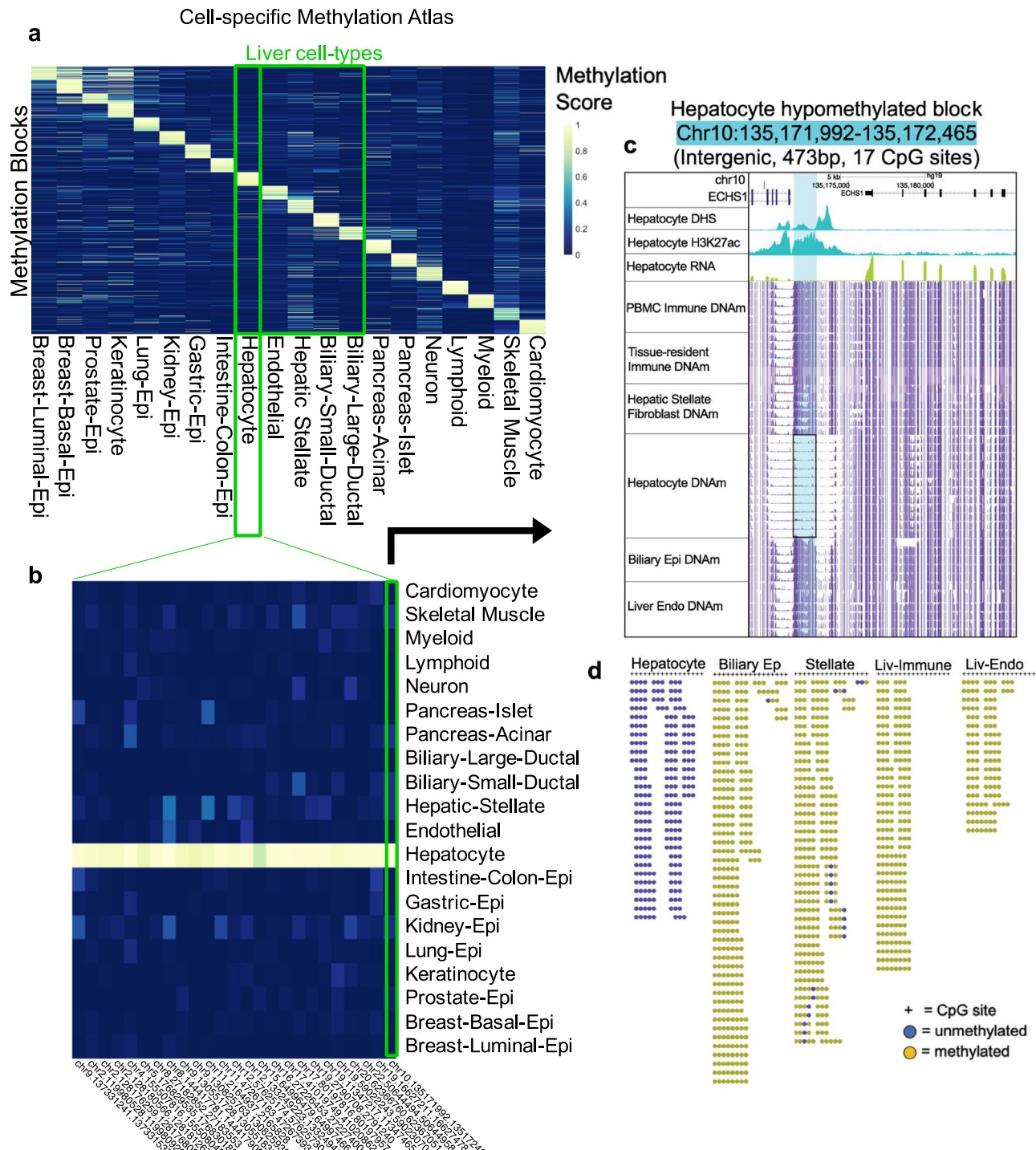

**Fig. 2 | Liver cell-type DNA methylation atlas relative to other healthy tissues.**
**a** Heatmap of differentially methylated, cell-type-specific blocks (DMBs) identified from reference WGBS data of healthy human cell-types. Each cell in the plot marks the methylation score of one genomic region (rows) at each of 20 cell-types (columns), with up to 50 blocks shown per cell type. The methylation score represents the number of fully unmethylated or methylated read-pairs divided by total coverage for hypo- and hyper-methylated blocks, respectively. **b** Heatmap highlighting the top 25 hepatocyte-specific DMBs. **c** Example of one hepatocyte-specific hypomethylated block (highlighted in blue), upstream of *ECHS1* highly expressed in hepatocytes (green track). The alignment from the UCSC genome browser depicts the average DNA methylation (DNAm, purple tracks) across WGBS samples from five different liver cell-types as well as PBMC samples. Chromatin organization marks in hepatocytes are displayed (blue tracks) to show accessibility (DNAse I hypersensitivity, DHS) and regulatory function (H3K27ac binding). **d** Fragment-level visualization of methylation sequencing reads at hepatocyte-specific hypo-methylated block in reference WGBS samples from five different liver cell-types.

additional chromatin accessibility and histone modification data to characterize the integrated epigenomes of hepatocyte, biliary epithelial, hepatic stellate, liver endothelial and liver-resident immune cell-types. We were encouraged to find that liver cell-type-specific hypomethylated

blocks were also regions with cell-type-specific chromatin accessibility and H3K27ac binding, emphasizing the regulatory importance of these regions in maintaining cell-type-specific features reflected in the multi-omic datasets (Fig. 3a, b).

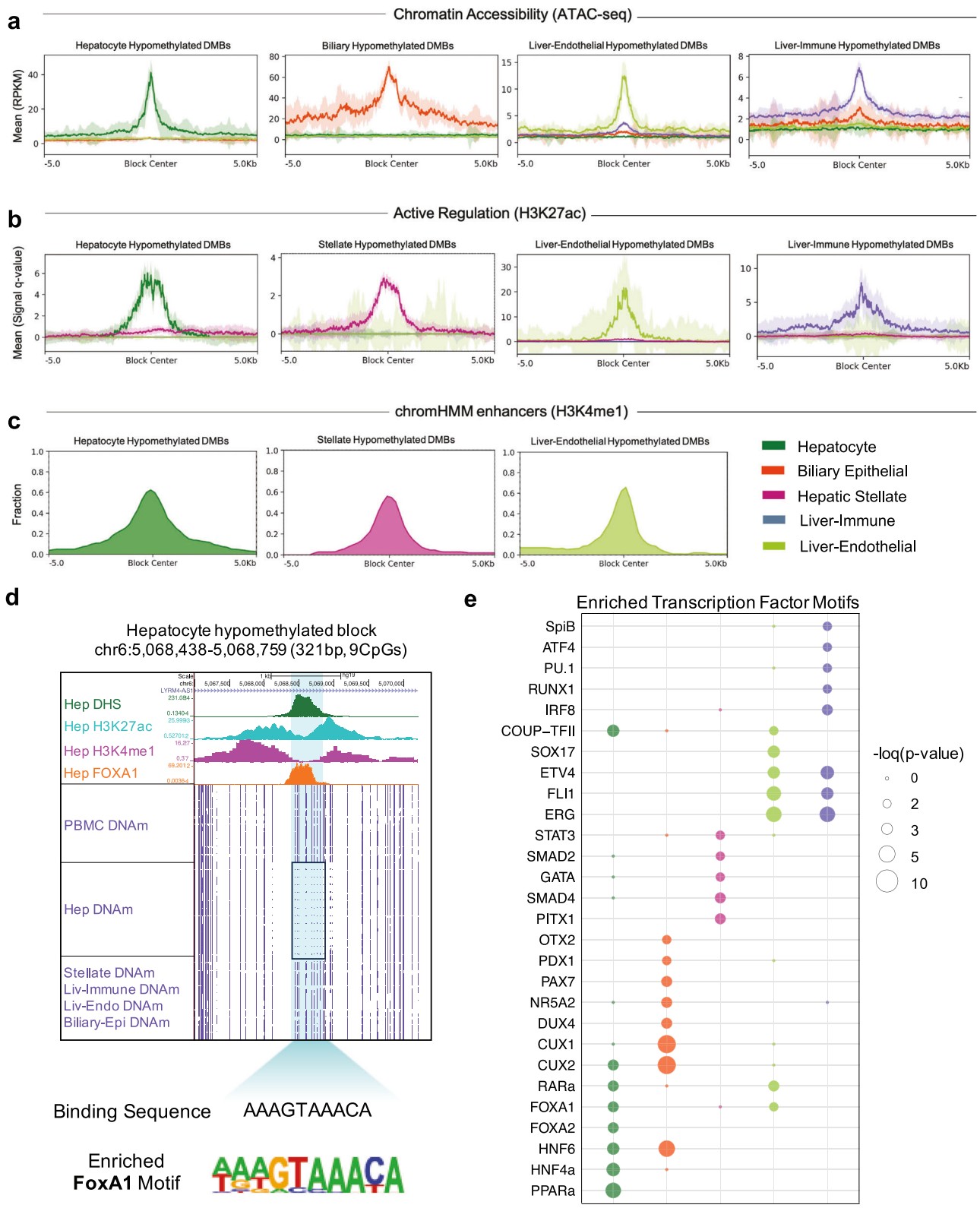

## Pioneer transcription factor binding sites (TFBS) enriched within liver cell-type-specific DNA methylation blocks

We performed motif analysis to explore association of the identified liver cell-type-specific DNA methylation with transcription factor binding. We found enriched motifs for several pioneer transcription factors within hypomethylated DMBs, including FOXA1/2, PAX7, CUX1, HNR5A2, DUX4, OTX2, GATA, SOX17, ATF4 and PU.1 (Fig. 3d, e). We

also found enriched motifs of binding sites for several liver developmental TFs known to cooperate with pioneer TFs, including HNF4a, HNF6, PDX1, RARa, COUP-TFII, and RUNX1[60–62]. Pioneer factors are a subclass of TFs that can bind to closed chromatin and elicit an extended functional capacity of the domain, often through local chromatin opening and demethylation[60]. As such, they act as master regulators of development and are known to drive cell fate

**Fig. 3 | Liver cell-specific hypomethylated DNA blocks coincide with other cell-specific epigenetic marks. a, b** Relationship of liver cell-type-specific hypomethylated blocks to chromatin accessibility and H3K27ac binding. Summary plots show the intensity of DHS/ATAC-seq or H3K27ac marks in a ±5-kb regions surrounding each hepatocyte-, biliary epithelial- or hepatic stellate-, liver endothelial-, and liver immune cell-specific hypomethylated block, respectively. Solid lines represent plot summary with standard error depicted by semi-transparent colored region. **c** Fraction of cell-type-specific hypomethylated blocks labeled as enhancers, associated with H3K4me1 mark, in chromHMM annotations for the same cell-types. **d** UCSC genome browser alignment at one example hepatocyte-specific hypomethylated block containing the FOXA1 binding sequence. Average methylation across WGBS samples shown in purple tracks. **e** Pioneer and developmental TF binding sites enriched within liver cell-type-specific hypomethylated blocks, from HOMER motif analysis using the findMotifsGenome.pl function. Shown are the binomial $p$-values (one-sided). Captured blocks without liver cell-type-specific methylation were used as background. Hepatocyte DHS, H3K27ac, and H3K4me1 data were obtained from the German Epigenome programme (DEEP). Hepatocyte FOXA1 Chip-seq data was obtained from the ENCODE project. Biliary epithelial ATAC-seq data and Hepatic stellate H3K27ac histone modification data were obtained from the ENCODE project. Source data are provided as a Source Data file.

transitions[60]. However, we were surprised to find binding sites for pioneer TFs also enriched within liver-specific hypermethylated DMBs (Supplementary Fig. 2c). Interestingly, CpG dinucleotides were also enriched within the motifs found in hypermethylated DMBs and several methylation-sensitive TFs were amongst the top hits, including NRF1 where methylation is known to directly repress TF binding (Supplementary Fig. 2b and Supplementary Data 5)[63]. Although less common, pioneer TFs have been shown to recruit transcriptional repressors and establish a closed and further silenced chromatin architecture[57,64,65]. Several TFBS of transcriptional repressors known to interact with pioneer TFs were also enriched, including TBET, TRPS1, ZNF669, and E2F7. Annotation of the majority of liver-specific hypermethylated DMBs to bivalent TSS/enhancer regions coincides with the ability of some pioneer TFs to simulate PRC2 complex-inducing H3K27me3-marked heterochromatin, often deposited on lineage-specific enhancers (Supplementary Fig. 2d)[66–68]. In composite, these results match with the role of the liver cell-type-specific methylation blocks identified here as being critical for cell identity.

## Origins of cellular damage immediately after liver transplant

To identify the origins of cfDNA fragments in the circulation of liver transplant patients, we used the top 100 methylation blocks for each cell-type group and generated an expanded liver cell-type-specific DNA methylation atlas (Methods). We then applied a fragment-level deconvolution algorithm, previously validated to estimate relative contributions from cfDNA methylation sequencing data (Supplementary Data 6a, 8a, and 9a)[35]. First, to explore the changing cfDNA makeup after liver transplant, we assessed changes across all patients comparing pre-transplant cfDNA origins to post-reperfusion changes in serially collected blood samples from 28 liver transplant patients on the day of surgery (POD0; Fig. 4a). We found that there was a significant ~5-fold increase in cfDNA concentration after transplant across all patients in the cohort, reflecting increased cell turnover from the surgical procedure itself ($p < 0.05$, Wilcoxon matched-pairs signed rank test) (Fig. 4f). From the deconvolution analysis we found that liver cell types mainly contributed to this increase (Fig. 4b) with a significant increase in hepatocyte, hepatic stellate, and endothelial cfDNA fraction and a corresponding relative decrease in myeloid cfDNA that constitutes most of the hematopoietic signal at baseline ($p < 0.05$, Wilcoxon matched-pairs signed rank test) (Fig. 4d, e, g, h). The concentration of hepatocyte cfDNA in genome equivalents/mL (Geq/mL) correlated with patient AST and ALT liver enzyme values (Spearman $r = 0.81$ AST and $r = 0.82$ ALT, $p < 0.05$) (Fig. 4c). The homogenous cfDNA changes across all patients show applicability of this approach across a diverse patient cohort. Beyond damages to the allografted liver cell-types, we found that the transplant procedure results in multiple tissue cellular damages of the recipient as well. We were surprised to find a significant increase in neuron-derived cfDNA after transplant (Fig. 4i). While only representing a small overall fraction of the total cfDNA, there was an average 4-fold increase in this signal indicating neuronal cell death during the procedure. In addition, there were also significant increases in cardiomyocyte, biliary-ductal and gastric-epithelial cfDNAs in Geq/mL (Supplementary Fig. 5).

## Sustained elevation of hepatocyte and biliary epithelial cfDNA indicate allograft injury

We collected additional serum samples in 20 liver transplant patients to explore cfDNA changes over time at defined timepoints during the first month after transplant, the highest risk period for post-transplant complications (Fig. 5a). Of these patients, 11 (55%) had liver biopsies showing allograft injury. For 5 of the 11 patients, these biopsies were within the first month overlapping with the timing of serum sample collection. For the remaining 6 patients the biopsies were within the first year after transplant. Patients with allograft injury were diagnosed from histopathological analysis of liver biopsy tissues. Liver biopsies are not typically considered the standard of care for routine monitoring of liver transplant patients[4]. Instead, they are usually performed only when clinically indicated. The majority of patients in this cohort designated as being "without allograft injury" did not have clinical indication for liver biopsies after transplant. Two patients were labeled "without injury" who had liver biopsies taken after transplant that were negative for pathology. These two borderline cases were both found negative for acute cellular rejection, using the standardized lesion grading, Banff Rejection Activity Index (RAI index ≤3). There were no differences in cfDNA concentration after transplant comparing these two outcome groups (Supplementary Fig. 6h) though we noticed changes in cfDNA composition when comparing across the entire cohort (Supplementary Fig. 6a). Liver-epithelial cellular damage mostly recovered in patients without allograft injury during the first post-operative week. In contrast, patients diagnosed with allograft injury during the first year after transplant had sustained elevation of hepatocyte and biliary epithelial cfDNA from POD7-POD30 ($p < 0.05$, Mann-Whitney test) (Fig. 5c–f). Despite these differences in liver epithelial signals, there was no significant difference in hepatic stellate or endothelial cfDNA associated with different outcomes (Supplementary Fig. 6c, d). These findings were irrespective of the type of allograft injury diagnosed at the eventual time of for-cause liver biopsy (FC-bx). Of the patients with allograft injury, 7 (of 20) were diagnosed with hepatocellular, 3 mixed hepatobiliary, and 1 biliary forms of allograft injury (Fig. 5b). Despite this variation, elevated liver epithelial cfDNA was detected during the first post-operative month in all 11 patients with allograft injury, with an elevated signal detected a median of 63 days [range 2-203 days] ahead of the time of tissue-biopsy based diagnosis. Comparing the trajectory of liver cell-type damages over time, patients without allograft injury had higher levels of lymphoid and endothelial cfDNA relative to hepatocyte and biliary epithelial cfDNA at POD30 (Fig. 5g). This suggests that the cell-specific composition of liver-derived cfDNA provides added context and that the ratio may be useful to monitor and predict injury patterns during the peri-transplant time period relative to the total liver-derived cfDNA signal (Supplementary Fig. 6e).

## Cell-free DNA methylation indicates the source of allograft injury

We added phenotype-matched samples from additional patients at the time of for-cause liver biopsy (FC-bx) used to diagnose allograft injury, evaluating 30 serum samples from 24 individuals (Fig. 6a). Samples

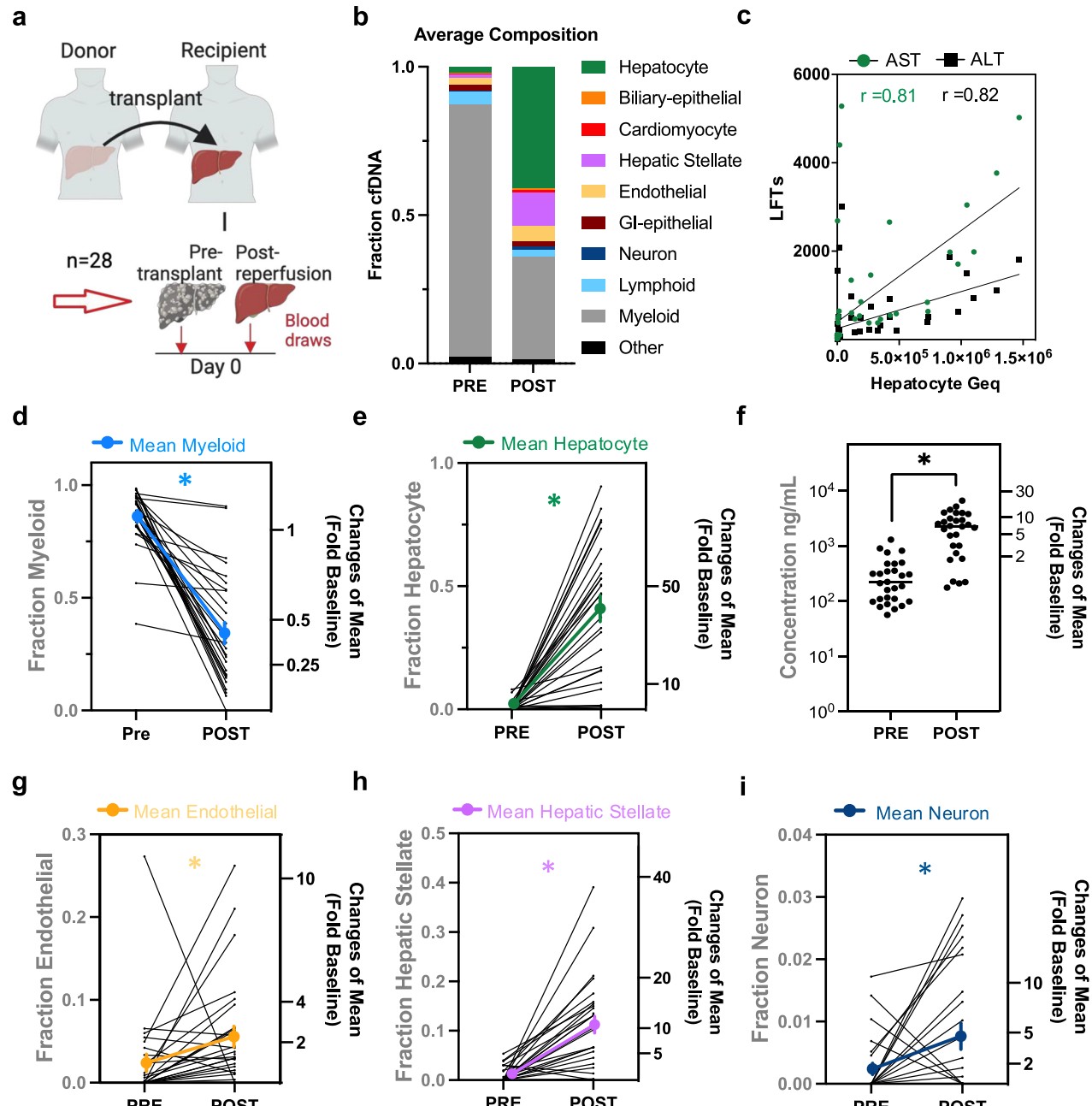

**Fig. 4 | Origins of cellular damage after liver transplant derived from methylation patterns of cfDNA fragments. a** Serial serum samples from 28 liver transplant patients collected pre-transplant and post-reperfusion on post-operative day 0 (POD0). **b** Average cellular origins of cfDNA estimates. **c** Correlation of AST and ALT enzyme activity with hepatocyte-derived cfDNA (Spearman r = 0.81 AST, r = 0.82 ALT, two-sided, p = 0.0021 AST, p = 0.004 ALT). Pre-transplant (PRE) and post-reperfusion (POST) fraction of cfDNAs from myeloid (**d**), hepatocyte (**e**), endothelial (**g**), hepatic stellate (**h**), and neuronal cells (**i**) of individual patients. The mean ± SEM of the cohort is shown in bold. Right axes: Fold change relative to pre-transplant. **f** Concentration of cfDNA isolated from patient serum. Individual patient and median values are shown. **d–f** Wilcoxon matched-pairs signed rank test was used for comparison amongst groups (two-sided, n = 28). NS p ≥ 0.05, *p < 0.05; myeloid p = 0.0001, hepatocyte p = 0.0001, endothelial p = 0.0025, hepatic stellate p = 0.0001, neuron p = 0.029, concentration p = 0.001. Source data are provided as a Source Data file. Figure 4a created in BioRender (2025) [https://BioRender.com/y41q857].

were classified as having hepatocellular (n = 14), biliary (n = 6), or mixed hepatobiliary (n = 10) forms of allograft injury from histopathological analysis of the paired biopsy tissues (see Supplementary Note 2 for full details). Notably, the composition of cfDNA was significantly different at the time of biopsy-proven phenotypes, comparing hepatocellular and biliary etiologies of allograft injury (Fig. 6b and Supplementary Data 6b, 8b, and 9b). Hepatocyte cfDNA was increased in samples with hepatocellular or mixed hepatobiliary injury

compared to pure biliary injury (p < 0.05, Mann–Whitney test) (Fig. 6c). Likewise, biliary cfDNA was increased in samples with biliary or mixed hepatobiliary injury compared to hepatocellular injury (p < 0.05, Mann–Whitney test) (Fig. 6d). There was also an increase in myeloid-derived cfDNA in patients with biliary etiologies of allograft injury relative to those with hepatocellular etiologies (p < 0.05, Mann–Whitney test) (Supplementary Fig. 7b–e). Further, there was an association between standardized lesion grading (Banff Rejection

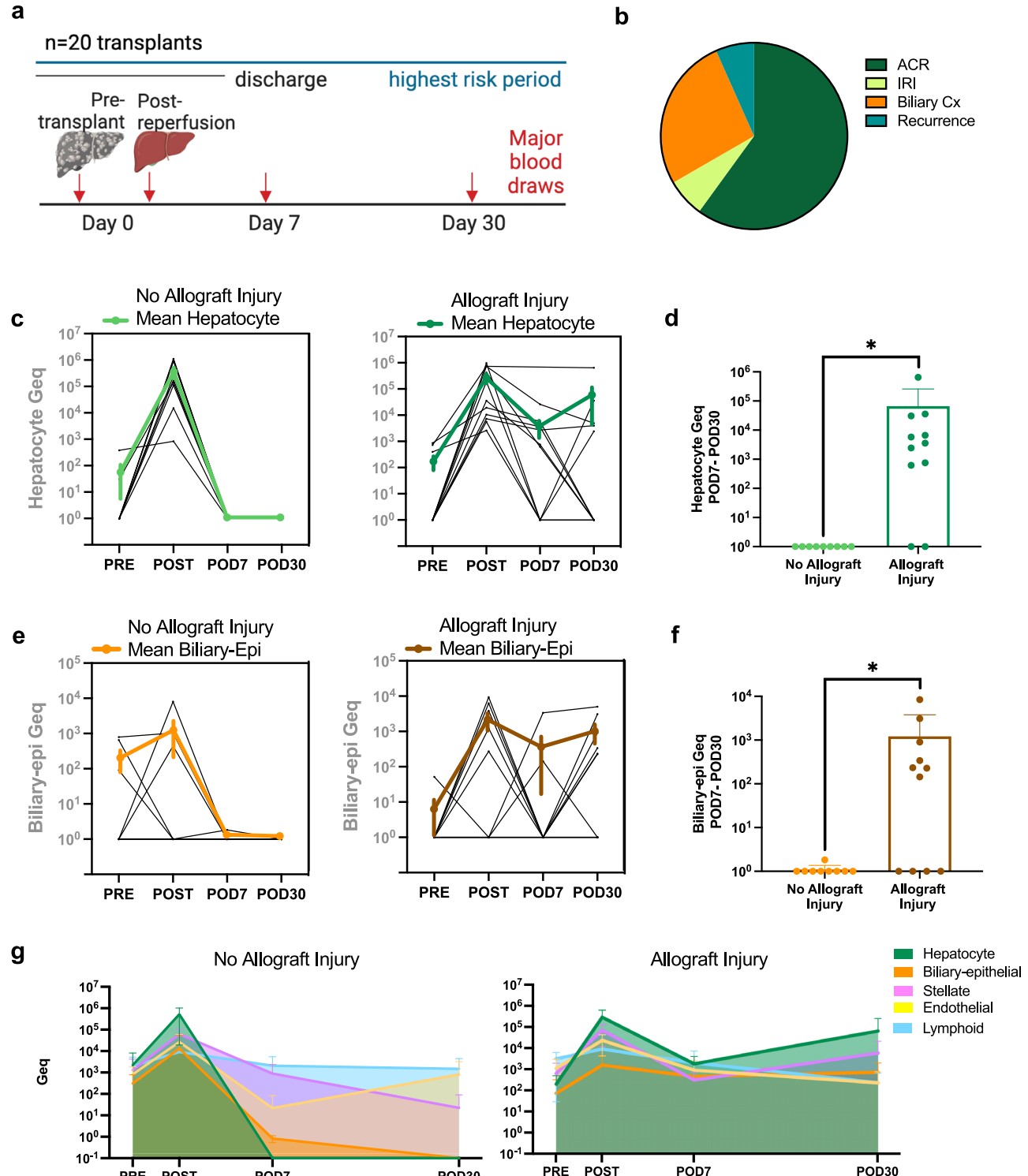

**Fig. 5 | Time course of cell-type-specific damages after liver transplant. a** Serial serum samples from 20 liver transplant patients collected pre-transplant, post-reperfusion (POD0), post-operative day 7 and 30 (POD7, POD30). **b** Etiologies of allograft injury. By 1 year after transplant, 11 of 20 patients were diagnosed with different etiologies of allograft injury by for-cause biopsy. **c** Hepatocyte cfDNA time course in patients with allograft injury (right, $n = 11$ patients) or no allograft injury (left, $n = 9$ patients). Mean ± SEM of each cohort is shown in bold. **d** Average combined hepatocyte cfDNA on POD7 and POD30 for each individual, grouped by outcome (allograft injury = 11 patients, no allograft injury = 9 patients). Bar plot displaying group Mean + SD (Mann–Whitney test, two-sided, $p = 0.006$). **e** Biliary

cfDNA time course in patients with allograft injury (right, $n = 11$ patients) or no allograft injury (left, $n = 9$ patients). Mean ± SEM of each cohort is shown in bold. **f** Average combined biliary cfDNA on POD7 and POD30, grouped by outcome (allograft injury = 11 patients, no allograft injury = 9 patients). Bar plot displaying group Mean + SD (Mann–Whitney test, two-sided, $p = 0.009$). **g** Time course of five liver cell type cfDNAs in patients with allograft injury (right, $n = 11$) or no allograft injury (left, $n = 9$). Mean ± SD of each cohort is shown. **d**, **f** NS $p \geq 0.05$, *$p < 0.05$. Source data are provided as a Source Data file. Figure 5a created in BioRender (2025) [https://BioRender.com/l14b192].

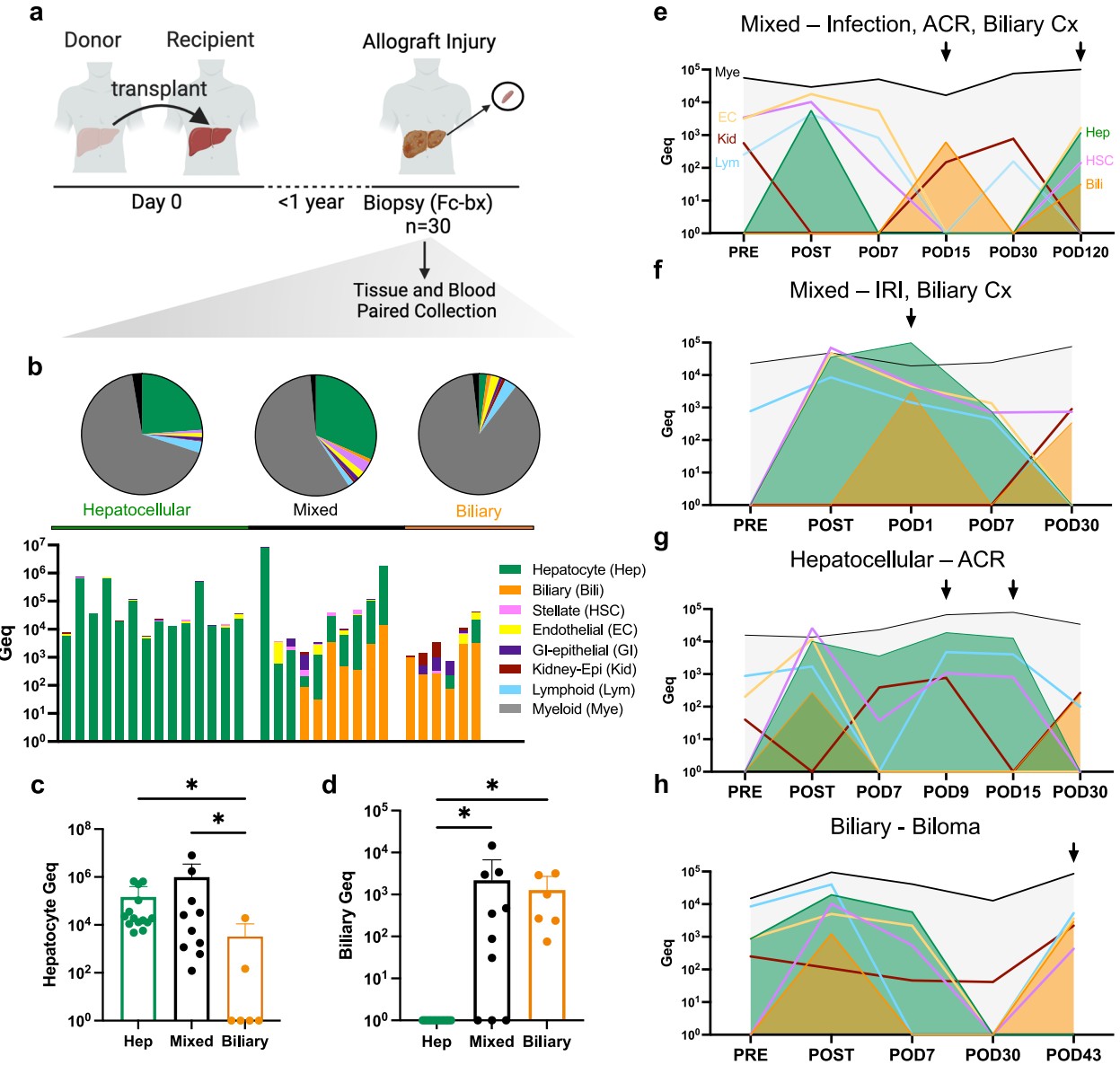

**Fig. 6 | Cell-free DNA methylation patterns indicate cellular sources of allograft injury. a** Serum samples collected at the time of for-cause liver biopsies (FC-bx) to diagnose allograft injury. All biopsies were taken within 1 year of liver transplant and samples are representative of 24 patients (*n* = 30 samples). **b** Cellular origins of cfDNAs classified by injury patterns observed in biopsies. Top, Average fractions of different cellular sources detected for each type of allograft injury. Bottom, Contingency stacked bar graph that depicts the proportion of solid-organ cellular Geq within each individual sample where each stack represents a different sample and the height of each segment within the stack represents the relative proportion of cfDNA within that cell-type group across samples. **c** Hepatocyte cfDNA in serum samples with hepatocellular or mixed hepatobiliary injury compared to biliary injury alone (Mann-Whitney test, two-sided, *p* = 0.003 Hep vs Biliary; *p* = 0.007 Mixed vs Biliary). Bar plot displaying group Mean + SD. **d** Biliary cfDNA in serum samples with biliary or mixed hepatobiliary injury compared to hepatocellular injury alone (Mann–Whitney test, two-sided *p* = 0.0001 Biliary vs Hep; *p* = 0.0003 Mixed vs Hep). Bar plot displaying group Mean + SD. **b**–**d** Serum samples classified as *n* = 14 hepatocellular, *n* = 10 mixed hepatobiliary, and *n* = 6 biliary etiologies of

allograft injury. **e**–**h** Time courses of cellular damage during the peri-transplant time period in patients with hepatocellular, biliary, and mixed hepatobiliary forms of allograft injury. Timepoints corresponding to complications and liver-biopsy proven diagnoses are marked by an arrow. **e** Patient with COVID-19 infection at POD15 and FC-bx diagnosis of acute cellular rejection (ACR) with hyperbilirubinemia at POD120 (mixed injury classification). Elevated kidney epithelial cfDNA detected on POD0, POD15, and POD30 match with the hepato-renal syndrome (HRS) diagnosis pre-transplant and acute kidney injury (AKI) after transplant. **f** Patient with FC-bx diagnosis of hepatic ischemia with hyperbilirubinemia at POD1 (mixed injury classification). AKI was indicated by elevated creatinine levels POD9 and elevated kidney epithelial cfDNA on POD30. **g** Patient with FC-bx diagnosis of ACR at POD9 and POD15 (hepatocellular injury classification). AKI was indicated by elevated creatinine levels on POD8 and elevated kidney epithelial cfDNA POD7, POD9, and POD30. **h** Patient with diagnosis of biloma at POD43 (biliary injury classification). Source data are provided as a Source Data file. Figure 6a created in BioRender (2025) [https://BioRender.com/r21k256].

Activity Index, RAI) of liver biopsy tissues and total liver cfDNA in serum samples of patients with allograft injury (Supplementary Fig. 6b). However, we did not find a difference in total cfDNA concentration comparing samples with different etiologies of injury (Supplementary Fig. 7c). The cfDNA composition changes over time

reflected the trajectory of cellular damages in patients with different injury types (Fig. 6e–h; patient details in the legend). At the time of FC-bx only hepatocyte cfDNA was detected in a patient with hepatocellular injury (Fig. 6g), only biliary epithelial cfDNA in a patient with pure biliary injury (Fig. 6h), and both hepatocyte and biliary epithelial

cfDNA in two patients with different etiologies of mixed hepatobiliary injury (Fig. 6e, f). Taken as a whole, distinct cellular damages after liver transplant are detectable by the analysis of blood samples.

## Discussion

The liver cell-type-specific DNA methylation atlas sheds light onto the epigenomic characteristics established early during development by identifying genomic regions of cell identity that are stably maintained in differentiated cells. We find that DNA methylation coincides with other liver cell-type-specific epigenetic marks, validating the biological relevance of these regions and enhancing their utility in detecting altered turnover of cells from DNA fragments shed into the circulation. Similar to other studies, we found the majority of liver-cell-specific DMBs to be hypomethylated. However, we found enriched TFBS of pioneer factors within both hypo- and hyper-methylated liver-cell-specific DMBs. Surprisingly, we also found CpG dinucleotides enriched within TFBS motifs associated with methylation-sensitive TFs in hypermethylated DMBs. This suggests that cell-type-specific hypo- and hyper- methylated regions may play a similar function in different contexts to repress precursor or stem cell transcriptional programs and control terminal differentiation into distinct cell types. Ultimately, this could serve as a valuable resource for many applications and shed light on factors needed for cell reprogramming that may also play a role in disease pathogenesis and cell-type-specific epigenetic regulation[69].

We identified sufficient numbers of DNA methylation blocks to discriminate cells of origin of liver-derived DNA fragments in the circulation, including hepatocyte, biliary-epithelial, hepatic stellate, and endothelial cells. As one limitation, we were unable to identify enough DMBs with sufficient specificity to profile liver-resident immune cell turnover in patient blood samples relative to all other cell-types included in the atlas. Likely this was due to the purity and extent of peripheral and tissue-resident immune cell methylome reference data available. Instead, extended liver-resident immune cell markers were identified with relaxed specificity thresholds to use for characterization of liver cell-specific epigenetic data (Supplementary Fig. 3, Supplementary Data 7). Generation of additional cell-specific methylation sequencing data to better characterize immune cell diversity will allow for enhanced ability to detect tissue-resident cell turnover in the future. In addition, implementation of hierarchical statistical models will allow for a more fine-tuned assessment of cell-free DNA composition in the face of limited numbers of highly specific methylation patterns to distinguish rare cell-types[70,71].

The expanded liver cell-type-specific methylation atlas allowed for detection of tissue-derived fragments in the circulation to reveal cell types in the recipient impacted by the transplant procedure, comparing post-reperfusion signals to the pre-transplant baseline. The donor liver can be damaged in several ways including, cold and warm ischemia, surgical anastomoses, and reperfusion injuries[1–3,33,34]. We found that these tissue effects were reflected by a relative increase in hepatocyte, hepatic stellate, and endothelial cfDNA compared to the myeloid-derived baseline signal. However, we were surprised by the increase in neuron and cardiomyocyte cfDNA after the transplant. The increase in neuron cfDNA could be caused by neurotoxicity from the general anesthesia; although, neurological complications are more common after liver (30%) than after heart (4%) or kidney transplants (0.5%)[72]. The neuron-specific methylation patterns used for analysis were identified using purified CNS neuron methylomes (not PNS neurons). This is also reflected by the functional characterization of these regions acting in programs relevant to CNS neurons (Supplementary Data 2 and Supplementary Fig. 4). Increased susceptibility or comorbidity due to the pathophysiology of the underlying hepatic disease may play a role and contribute to increased neuronal cell death in liver transplant patients. Interestingly, the concentration of the pre-transplant cfDNA was ~100-fold higher compared to cfDNA from

healthy individuals. Inflammation and activation of coagulation in liver failure has been found to increase cfDNA concentration, but also the increased concentration could result from impaired clearance mechanisms since these individuals all have end-stage liver disease and many also have kidney failure as a co-morbidity. Other studies have also found elevated cfDNA concentrations in individuals with impaired liver tissue function as well as demonstrated similar concentrations after solid organ transplants[25,29,73,74].

The large increase in liver-cell-derived cfDNA after transplant can serve as a proof-of-concept to validate the prediction accuracy of deconvolution results. We used a complete fragment-level deconvolution model to estimate relative abundance and changes in the cfDNA composition, shown to accurately detect cfDNA from a source at 0.1% resolution[35]. Also, we found a significant correlation between hepatocyte cfDNA and AST/ALT liver enzyme activity (Fig. 4c). However, we did not find biliary cfDNA to correlate significantly with alkaline phosphatase (ALP) or bilirubin levels (Supplementary Fig. 6f, g). The short half-life of cfDNA (15 min–2 h) relative to commonly monitored liver function parameters also contributes to some discrepancy between these parameters[75,76]. Changes in cell-type-specific cfDNAs reflect changes in cell turnover and thus measure different facets of tissue dysfunction. It is noteworthy that hepatocyte cfDNA is typically detectable in healthy individuals whereas biliary-epithelial cfDNA is mostly below detection (Supplementary Fig. 10g). As a major limitation, the cfDNA used in this study was extracted from serial serum samples. Therefore, we performed size-selection of cfDNA to reduce contaminating HMW-DNA from hematopoietic cell lysis during the process of serum preparation to allow for generalizability to other analyses of banked serum samples. The analysis of paired plasma and serum samples demonstrates strong correlation for the solid organ-derived cfDNAs from either source (Supplementary Figs. 8–10) (See Supplementary Note 1 for further discussion). However, we cannot exclude other differences impacting preanalytical factors that make it challenging to compare cfDNA isolated from serum and EDTA-plasma. After size selection, we found the concentration of cfDNA in serum samples to be ~2 times higher than in paired plasma samples at the same timepoint. In the future, we recommend use of specialized blood collection tubes containing stabilizing preservatives to prevent cell lysis for consistent results across multi-center studies.

We found that sustained elevation of liver epithelial cfDNA during the first month after transplant was associated with allograft injury, while patients without allograft injury had significantly reduced levels of liver epithelial cfDNA as early as the first week post-transplant. Importantly, liver cell-specific methylation patterns appear to be stably maintained during the ongoing processes of tissue damage, repair, and remodeling after transplant. We were able to detect elevated liver epithelial signals in all patients with allograft injury, despite being a diverse cohort with several different types of allograft injury represented. Our results suggest that cell-free methylated DNA has predictive and diagnostic value to detect allograft injury earlier than clinical diagnosis by liver biopsy. Notably, we did not find a significant difference in standard biochemical indicators of tissue injury in patients with allograft injury (Supplementary Fig. 12). We also did not find a significant difference in hepatic stellate or endothelial cfDNA comparing patients with allograft injury to those without allograft injury during the first month post-transplant. This observation is in contrast with liver damage after radiation treatment of patients with right-sided breast cancers where liver endothelial cfDNA showed a >10-fold increase after radiation and delayed recovery to baseline one month after treatment in comparison to hepatocyte cfDNA[38]. Thus, cfDNAs reflect distinct cellular responses to different types of injury and repair in the same organ.

Beyond damage to the allograft, altered cfDNA methylation patterns are also able to reveal cellular damages of other patient organs to indicate extra-hepatic toxicity and immune cell turnover[23,30,31]. This is a

useful application of cfDNA methylation patterns that can simultaneously allow for monitoring of common pulmonary, renal, cardiac, and neurological complications after liver transplants. Acute kidney injury (AKI) is one of the most common post-operative complications, occurring in up to 78% of liver transplant patients[2]. We were able to detect elevated kidney epithelial cfDNA in several patients in our cohort experiencing hepato-renal syndrome (HRS) pre-transplant as well as those experiencing AKI post-transplant (Fig. 6e–h). In addition, we noticed divergent trajectories of lymphoid versus myeloid cfDNA, with lymphoid cfDNA demonstrating more dynamic changes compared to myeloid cfDNA that remains a constant background signal (Fig. 6e–h and Supplementary Fig. 7). We found elevated lymphoid cfDNA corresponding to infection in several patients, including one patient with a COVID-19 infection (Fig. 6e).

Many studies have demonstrated the utility of donor-derived (dd) cfDNA to detect allograft injury[12,14–17]. However, dd-cfDNA is unable to discriminate amongst different causes of allograft injury. Likewise, it remains a challenge to distinguish causes relying on clinical presentation alone. Therefore, liver biopsy is still the gold standard to confirm a diagnosis and evaluate for response to treatment[4]. Here we found that cfDNA methylation is able to detect and differentiate hepatocellular versus biliary causes of allograft injury at the time of biopsy-proven diagnosis (FC-bx). Biliary complications after liver transplant, such as ascending cholangitis, strictures (both anastomotic and non-anastomotic), leaks, and recurrence of primary sclerosing cholangitis, contribute significantly to post-transplant morbidity and mortality in both living and deceased donor transplant recipients. Conventional diagnostic and monitoring methods for these conditions often necessitate cross-sectional imaging techniques, such as MRCP, or invasive procedures like ERCP or liver biopsy, which pose additional risks to patients[77,78]. Enhanced detection of biliary cell-type-specific damage allows for differentiation from hepatocellular forms of allograft injury and associated tissue damage. This enables an earlier and more accurate diagnosis of biliary complications and improved non-invasive monitoring post-treatment. Incorporating cfDNA as a diagnostic tool into clinical practice could potentially reduce the need for invasive procedures and facilitate early intervention with targeted treatment. However, only samples having tissue-biopsy proven etiologies of allograft injury and paired serum and tissues samples collected at the same time were used in this study. Additional large-scale studies are needed to fully encompass the spectrum of possible injuries and also to fully elucidate the capacity of cfDNA methylation patterns to predict and diagnose injury earlier than existing biochemical markers of injury.

In summary, we show that cfDNA methylation patterns found on fragments released from dying cells can indicate the origins of cell death and tissue damage in transplant patients. Expanded atlases of DNA methylation sequencing data allow for identification of cfDNA fragments originating from a variety of cell-types in the liver, demonstrating applicability in a wide range of clinical settings. We correlate our findings from the cfDNA methylation analysis with clinical data, histopathological results, and outcomes of conventional clinical monitoring. We conclude that cell-free methylated DNA in the circulation of liver transplant patients can indicate allograft injury and discriminate amongst causes of allograft injury matching with tissue biopsy-proven diagnosis.

## Methods
### Study approval
Liver transplant patients were enrolled and provided signed informed consent in this Georgetown University Medical Center and MedStar Georgetown University Hospital IRB-approved study (IRB protocols # IRB 2017-0690 and # 2017-0365).

### Study cohort
Serial serum samples were collected from 28 liver transplant patients at predetermined timepoints: pre-transplant (PRE) and post-reperfusion (POST) on post-operative day 0 (POD0), post-operative day 7 (POD0), and post-operative day 30 (POD30). Beyond this, we also collected samples in patients experiencing complications at the time of symptom presentation. Further, we added phenotype-matched samples from an additional 16 patients at the time of for-cause liver biopsy (FC-bx) used to diagnose allograft injury. Samples were classified as having hepatocellular ($n = 14$), biliary ($n = 6$), or mixed hepato-biliary ($n = 10$) forms of allograft injury from histopathological analysis of paired liver biopsy tissues (see Supplementary Note 2 for additional details). A schematic of the time series for sample collection can be found in Fig. 1.

### Isolation of serum samples
Peripheral blood (~6–12 mL) was collected in red-top venous puncture tubes and allowed to clot at room temperature for 30 min before centrifugation at $1200 \times g$ for 10 min at room temperature to separate the serum fraction. Patient characteristics with samples analyzed in this study are summarized in Supplementary Data 1. Self-reported sex was collected for all participants consenting to share this information. In order to maintain relevant sample sizes for analysis, statistical analyses were performed on males/females together. However, all sex data is made available in the source data. No gender-related data was collected.

### Isolation of plasma samples
Peripheral blood (~6–9 mL) was collected in lavender-top venous puncture tubes (EDTA tubes). Plasma was collected as a part of the Ficoll-Paque density gradient separation method to isolate Peripheral Blood Mononuclear Cells (PBMCs), centrifuged at $1000 \times g$ for 20 min at room temperature without brakes. The separate plasma fraction was then centrifuged again at $3000 \times g$ for 10 min before cell-free DNA (cfDNA) isolation.

### Isolation of circulating cell-free DNA (cfDNA)
Circulating cell-free DNA was extracted from human serum or plasma, using the QIAamp Circulating Nucleic Acid kit (Qiagen) according to the manufacturer's instructions. Cell-free DNA was quantified via Qubit fluorometer using the dsDNA BR Assay Kit (Thermo Fisher Scientific). Additional size selection using Beckman Coulter beads was applied to remove high-molecular weight DNA reflective of cell-lysis and leukocyte contamination[38,79]. Paired serum and plasma were processed at serial timepoints for $n = 3$ liver transplant patients and $n = 4$ healthy controls to serve as a quality control (Supplementary Figs. 8, 9, and 10) (See Supplementary Note 1 for further discussion). Fragment size distribution of isolated cfDNA after size selection was validated on the 2100 Bioanalyzer TapeStation (Agilent Technologies).

### Cell isolation to generate reference liver cell-type epigenomes
Reference epigenomes were generated for human liver cell types to expand upon publicly available datasets (Supplementary Data 2). Human biliary tissues were obtained from organs not suitable for transplant that were otherwise normal according to surgical assessment. Tissues were dissected, with samples processed from lobes, common hepatic duct, gallbladder, and common bile duct. Biliary epithelial cells (EpCAM+) were isolated from the dissected tissues according to previously established protocols with some modifications (Supplementary Fig. 1b)[80,81]. The inner epithelial layer was separated from the outer fibrous connective tissue and muscle layer using disposable sterile scalpels and dissecting scissors. The inner epithelial layer was then minced and enzymatically digested in a solution of Collagenase (2 mg/mL, Roche)- DNase (0.5%,ThermoFisher)- Fetal Bovine Serum (2% FBS, GIBCO) in Advanced DMEM/F-12 (Invitrogen)

supplemented with 1% Penicillin/Streptomycin (Pen/Strep, GIBCO) and 1% L-Glutamine (GIBCO). Using a set 1 h program (37C_h_TDK3) on the GentleMACS Octo Dissociator (Miltenyi Biotec), tissues were further digested by transferring solution into gentleMACS C-tubes (Miltenyi Biotec). After digestion, samples were passed through a 70 μm cell strainer (Corning) and washed with DMEM/F-12 (Invitrogen). Samples were centrifuged at 300xg for 10 mins and the supernatant removed. Cells from intrahepatic ducts (lobes) were underlayed with an equal volume of 20% and 50% (v/v) Percoll (Sigma). Following centrifugation at 1800 × g for 30 min at 4 °C, the intrahepatic biliary epithelial (IHBEC) fraction at the interface of the 20% and 50% Percoll layers was collected, washed and re-suspended in EasySep bead buffer (PBS, 0.5% BSA, 0.5 M EDTA). Biliary epithelial cells (EpCAM+) were isolated from pelleted cells using EasySep magnetic beads for EpCAM positive selection (StemCell Technologies Cat #17846) according to the manufacturer's protocol. Retained cells were considered EpCAM+ biliary epithelial cells. Efficiency of isolation was determined using flow cytometry (FITC Anti-human Epithelial cell, clone 5E11.3.1; StemCell Technologies Cat #60147Fl). Validation of biliary epithelial cell purity was done using DNA methylation at previously published regions with specificity to biliary tissues (Supplementary Fig. 11c). Cryopreserved passage 1 human liver sinusoidal endothelial cells (LSEC) were purchased from ScienCell research laboratories (SKU#5000). Cryopreserved passage 0 liver-resident immune cells and passage 1 human hepatic stellate cells were isolated from single donor healthy human tissues purchased from Novabiosis Lot: QGJ and JNA (liver-immune); Lot: ZMC and WAP (hepatic-stellate). Paired RNA-seq data was generated from the same cell-populations used for DNA methylation profiling to validate the identity of cell-types obtained from commercial sources through analysis of cell type expression markers (Supplementary Fig. 11).

### Isolation and fragmentation of genomic DNA

Genomic DNA from tissues was extracted with the DNeasy Blood and Tissue Kit (Qiagen) following the manufacturer's instructions and quantified via the Qubit fluorometer dsDNA BR Assay Kit (Thermo Fisher Scientific). Genomic DNA was fragmented via sonication using a Covaris M220 instrument to the recommended 150–200 base pairs before library preparation. Lambda phage DNA (Promega Corporation) was also fragmented and included as a spike-in to all DNA samples at 0.5%w/w, serving as an internal unmethylated control. Bisulfite conversion efficiency was calculated through assessing the number of unconverted C's on unmethylated lambda phage DNA.

### Bisulfite capture-sequencing library preparation

Bisulfite capture-sequencing libraries were generated according to published protocols. In brief, WGBS libraries were generated using the Pico Methyl-Seq Library Prep Kit (Zymo Research) according to the manufacturer's instructions. Library quality control was performed with an Agilent 2100 Bioanalyzer and quantity determined via the KAPA Library Quantification Kit (KAPA Biosystems). WGBS libraries were then pooled to meet the required 1 μg DNA input necessary for targeted enrichment. However, no more than four WGBS libraries were pooled in a single hybridization reaction and the 1 μg input DNA was divided evenly between the libraries to be multiplexed. Hybridization capture was carried out according to the SeqCap Epi Enrichment System protocol (Roche NimbleGen, Inc.) using SeqCap Epi CpGiant probe pools with xGen Universal Blocker-TS Mix (Integrated DNA Technologies, USA) as the blocking reagent. Washing and recovering of the captured library, as well as PCR amplification and final purification, were carried out as recommended by the manufacturer. The capture library products were assessed by Agilent Bioanalyzer DNA 1000 assays (Agilent Technologies, Inc.). Bisulfite capture-sequencing libraries with inclusion of 15–20% spike-in PhiX Control v3 library (Illumina) were clustered on an Illumina Novaseq 6000 S4 flow cell followed by 150 bp paired-end sequencing.

### Bisulfite sequencing data alignment and preprocessing

Paired-end FASTQ files were trimmed using TrimGalore (V 0.6.6)[82] with parameters "--paired -q 20 --clip_R1 10 --clip_R2 10 --three_prime_clip_R1 10 --three_prime_clip_R2 10". Trimmed paired-end FASTQ reads were mapped to the human genome (GRCh37/hg19 build) using Bismark (V 0.22.3)[83] with parameters "--non-directional", then converted to BAM files using Samtools (V 1.12)[84]. BAM files were sorted and indexed using Samtools (V1.12). Reads were stripped from non-CpG nucleotides and converted to BETA and PAT files using *wgbstools* (V 0.1.0) (https://github.com/nloyfer/wgbs_tools), a tool suite for working with WGBS data while preserving read-specific intrinsic dependencies[85]. The BETA files (a wgbstools-compatible binary format) contain position and average methylation information for single CpG sites. The PAT files contain fragment-level information (including CpG starting index, methylation pattern of all covered CpGs and number of fragments with exact multiCpG pattern).

### Reference DNA methylation data from healthy tissues and cells

Availability of previously published and publicly available WGBS data from healthy cell-types and tissues used in this paper are described in Supplementary Data 2. Controlled access to reference WGBS data from normal human tissues and cell types were requested from public consortia participating in the International Human Epigenome Consortium (IHEC)[86] and upon approval downloaded from the European Genome-Phenome Archive (EGA), Japanese Genotype-phenotype Archive (JGA), database of Genotypes and Phenotypes (dbGAP), and ENCODE portal data repositories[87]. Reference WGBS data were also downloaded from selected GEO and SRA datasets. Reference WGBS data were analyzed as previously described[38].

### Generation of expanded cell-type-specific DNA methylation atlas

Previously established atlases of cell-type-specific DNA methylation were refined to include expanded data generated from liver cell-types and curated from recently published WGBS dataset of purified healthy human cell-types (Supplementary Data 2). Tissue and cell-type-specific methylation blocks were identified from reference WGBS (see Supplementary Note 3 for full details)[38]. In brief, data was first segmented into blocks of homogenous methylation and then analysis was restricted to blocks covered by the hybridization capture panel used in the analysis of cfDNA (probed regions span 80 Mb (~20% of CpGs) on the capture panel)[35]. We also restricted analysis to blocks containing a minimum of three CpG sites, with lengths less than 2 kb and at least 10 observations. Samples were divided into 20 groups by cell-type and we performed a one-vs-all comparison to identify differentially methylated blocks unique for each group. For this we used the find_markers Rscript (with parameters "--tg.quant 0.2 --bg.quant 0.1 --margin 0.4") to calculate the average methylation per block/sample and rank the blocks according to the difference in average methylation between any sample from the target group and all other samples (https://github.com/nloyfer/MarkovDeconv)[38]. Blocks with a (−) direction are hypomethylated and (+) direction are hypermethylated, defined as a direction of methylation in the target cell-type relative to all other tissues and cell-types included in the atlas. Identified liver cell-type-specific DMBs meeting these specified requirements can be found in Supplementary Data 3. Extended cell-type-specific blocks for liver-resident immune cells can be found in Supplementary Data 7.

### Methylation score and visualization of cell-type-specific methylation atlas

Each DNA fragment was characterized as U (mostly unmethylated), M (mostly methylated) or X (mixed) based on the fraction of methylated

CpG sites as previously described[29]. We used thresholds of ≤33% methylated CpGs for U reads and ≥66% methylated CpGs for M. We then calculated a methylation score for each identified cell-type-specific block based on the proportion of U/X/M reads among all reads. The U proportion was used to define hypomethylated blocks and the M proportion was used to define hypermethylated blocks. Heatmaps were generated using the pretty heatmap function in the RStudio Package for the R bioconductor (RStudioTeam, 2015).

## UXM fragment-level deconvolution

An expanded atlas was generated using the top 100 methylation blocks per cell-type group using the 'uxm build' function with parameters "--rlen 3" from the UXM_deconv repository (https://github.com/nloyfer/UXM_deconv) (expanded atlas available at GEO under accession no. GSE262275)[35]. From this, each fragment is annotated as U (mostly unmethylated), M (mostly methylated) or X (mixed) depending on the number of methylated and unmethylated CpG sites. For each DMB, the proportion of U/X/M fragments is calculated across all reference WGBS cell-types and the U/M proportion reported for hypomethylated and hypermethylated DMBs, respectively. Then, the cell-type origins of cell-free DNA fragments isolated from serum of liver transplant patients are estimated using the 'uxm deconv' function with parameters "--rlen 3". Briefly, a non-negative least squares (NNLS) algorithm is used to fit each cell-free DNA sample and estimate its relative contributions. Predicted cell-type proportions were converted to genome equivalents and reported as Geq/mL through multiplying the relative fraction of cell-type-specific cfDNA times the concentration of cfDNA (ng/mL) by the mass of the human haploid genome $3.3 \times 10^{-12}$ grams.

## Chromatin accessibility and histone modification data generation and analysis

ATAC-seq libraries were generated from liver sinusoidal endothelial (LSEC) and liver-resident immune cryopreserved cells using the ATAC-seq kit (Active Motif). H3K27ac histone modification data was generated from liver sinusoidal endothelial (LSEC) and liver-resident immune cryopreserved cells using the Cut&Tag-IT Assay kit (Active Motif, H3K27ac antibody cat#39133). Library products were assessed by Agilent Bioanalyzer HS DNA assays (Agilent Technologies, Inc.) and clustered on an Illumina Novaseq 6000 S4 flow cell followed by 150 bp paired-end sequencing with inclusion of 10–15% spike-in PhiX Control v3 library (Illumina). Controlled access to DNase-seq, H3K4me1 and H3K27ac ChIP-seq data from hepatocytes was obtained from the German Epigenomme Program (DEEP) (EGAD00001002527) and publicly available ATAC-seq data from biliary tissue (gallbladder) was downloaded from the ENCODE project (ENCSR695FLC). Imputed data on H3K27ac binding in hepatic stellate cells was downloaded from the ENCODE project in bigWig format (ENCSR225UAF). ChromHMM annotations (18-state version) were downloaded from the ENCODE project (hepatocyte: ENCSR075JST, hepatic stellate: ENCSR593ZNP, endothelial: ENCSR227ZSK, neuron: ENCSR539JGB) and genomic regions associated with H3K4me1 enhancer mark were extracted and reformatted in bigWig format. Paired-end FASTQ files were trimmed using TrimGalore (V 0.6.6) with parameters "--paired -q 20". Trimmed paired-end FASTQ reads were mapped to the human genome (GRCh37/hg19 build) using Bowtie2-align (V 2.3.5.1)[88] with parameters "-X 1000 --local --very-sensitive --no-mixed --dovetail --phred33". Duplicated reads were marked with Picard (V 2.18.14) and reads with low mapping quality, duplicated or not mapped in proper pair were excluded using Samtools view (V 1.12)[84] with parameters "-F 1796 -q 30". BAM files were sorted and indexed using Samtools. Chromatin accessibility data was normalized to the same depth and then bigWig files were created using deepTools (V 3.5.1)[89] with the functions 'multiBamSummary ' and 'bamCoverage' using parameters "--normalizeUsing RPKM --binSize 25". Detection q-values were calculated for histone modification data relative to Input control using the macs3 (V3.0.0a6)[90] bdgcmp function (-m qpois). This method uses the BH process for poisson p-values to calculate the score in any bin using control (Input) as lambda and treatment (IP sample) as observation. BigWig files were then generated using bedGraphToBigWig. Summary plots were prepared using deepTools (V 3.5.1) with functions 'computeMatrix' and 'plotProfile' using default parameters, except for 'referencePoint = center' and 5 Kb margins.

## Functional annotation, transcription factor binding site and pathway analysis

Cell-type-specific methylation blocks were annotated using HOMER annotatePeaks.pl function (V4.11.1)[91]. Transcription factor binding site analysis was performed using the HOMER findMotifsGenome.pl function for known and de novo motifs with parameters "-mask -chopify -size given -cpg" and with captured blocks without liver cell-type-specific methylation used as background. All motifs with binomial p-value < 0.05 were considered. Pathway analysis of genes adjacent to identified tissue and cell-type-specific methylation blocks was performed using Ingenuity Pathway Analysis (IPA) (Qiagen)[92].

## Genome browser and fragment-level visualizations

Reference WGBS samples were uploaded as custom tracks for visualization on the UCSC genome browser[93]. Methylomes were converted to bigWig format using the wgbstools beta2bw function. Fragment-level visualization of methylation sequencing reads was performed with the wgbstools vis function with parameters "−min 3 −yebl --strict". Chromatin accessibility, histone modification, and RNA expression data were downloaded from the IHEC data portal as bigWig files (hg19). FOXA1 ChIP-seq data in liver was downloaded from the ENCODE project (ENCSR735KEY). Samples of the same cell type were averaged using multiBigwigSummary (v.3.5.1).

## RNA isolation and RNA-sequencing analysis

RNA was isolated from sorted cells using the RNeasy Kit (Qiagen) according to the manufacturer's protocol and quantified by Qubit RNA BR assay (Thermo Fisher Scientific). Total RNA was validated using an Agilent RNA 6000 nano assay on the 2100 Bioanalyzer TapeStation (Agilent Technologies). The resulting RNA Integrity number (RIN) of samples selected for RNAseq analysis was at least 7. RNA-sequencing libraries were prepared using TruSeq Total RNA library Prep Kit (Illumina) at Novogene Corporation Inc., and 150 bp paired-end sequencing was performed on an Illumina HiSeq 4000 with a depth of 50 million reads per sample. A reference index was generated using GTF annotation from GENCODEv28. Raw FASTQ files were aligned and assembled to GRCh38 and GRCh37 with HISAT2 / Stringtie (V 2.1.0). The differential expression was analyzed in R with packages EdgeR (V 3.32.1) and Rsubread (V1.6.3). Derived counts per million and p-values were used to create a rank ordered list, which was then used for subsequent integrative analysis. Expression levels at known cell type markers from single cell expression databases were used to validate the identity of isolated cell type populations for methylome analysis (Supplementary Fig. 11).

## Statistics

Statistical analyses for group comparisons and correlations were performed using Prism (GraphPad Software, Inc., United States) and R (V 4.1.3). A correlation analysis was performed to assess relationship between changing cell-free methylated DNAs and LFTs using Spearman's Rank Correlation Coefficient. Statistically significant comparisons are shown, with significance defined as $p < 0.05$. Correction for multiple hypothesis comparisons was performed using the Benjamini−Hochberg (B-H) method corrected p-value to control the

false discovery rate (FDR) from multiple pathways being tested against each gene-set. A two-stage linear step-up procedure of Benjamini, Krieger and Yekutieli was performed for *p*-value adjustment from multiple outcome measures.

**Reporting summary**

Further information on research design is available in the Nature Portfolio Reporting Summary linked to this article.

## Data availability

The DNA methylation data generated in this study (in BETA and PAT file formats) have been deposited in the GEO database under accession code GSE262275. Chromatin accessibility (ATAC-seq) and H3K27ac histone modification data generated in this study (in bigwig file format) are available in the GEO database under accession code GSE263243. Raw sequencing files for all data generated in this study are available at the database of Genotypes and Phenotypes (dbGAP) [https://www.ncbi.nlm.nih.gov/gap/] under accession code phs003610.v1.p1 [http://www.ncbi.nlm.nih.gov/projects/gap/cgi-bin/study.cgi?study_id=phs003610.v1.p1] as controlled access for patient privacy and consent purposes. Access can be obtained from the Data Access Committee (DAC) after a Data Use Certification (DUC) is submitted and reviewed for compliance with relevant policies and data use limitations. The reference, publicly available WGBS data from healthy cell-types and tissues used in this study are available in the dbGAP database under accession code phs003290.v1.p1 [https://www.ncbi.nlm.nih.gov/projects/gap/cgi-bin/study.cgi?study_id=phs003290.v1.p1]; the ENA database [https://www.ebi.ac.uk/ena/browser/home] under accession codes: PRJNA342657, PRJNA353755, PRJNA430908, PRJNA596240; the ENCODE database [https://www.encodeproject.org/] under accession codes: ENCSR116JEF, ENCSR267SNS, ENCSR577RUU, ENCSR656TQD, ENCSR999RWT; the AMED-CREST [https://crest-ihec.jp/english/database/index.html] partnered with IHEC database [https://epigenomesportal.ca/ihec/index.html] registered by DDBJ and operated by NBDC [https://humandbs.dbcls.jp/en/] under accession codes: JGAX00000006927, JGAX00000006928, JGAX00000006929, JGAX00000006930, JGAX00000006932, JGAX00000006933, JGAX00000006934, JGAX00000011784, JGAX00000011793, JGAX00000011802, JGAX00000011811, JGAX00000012144, JGAX00000012145, JGAX00000012146, JGAX00000012147, JGAX00000012148, JGAX00000012149, JGAX00000012150, JGAX00000012151, JGAX00000012152, JGAX00000012153, JGAX00000012154, JGAX00000012155; and the EGA database [https://ega-archive.org/] under accession codes: EGAD00001002281, EGAD00001002294, EGAD00001002305, EGAD00001002309, EGAD00001002325, EGAD00001002346, EGAD00001002354, EGAD00001002361, EGAD00001002364, EGAD00001002367, EGAD00001002370, EGAD00001002383, EGAD00001002395, EGAD00001002403, EGAD00001002416, EGAD00001002423, EGAD00001002429, EGAD00001002460, EGAD00001002464, EGAD00001002483, EGAD00001002486, EGAD00001002492, EGAD00001002496, EGAD00001002501, EGAD00001002508, EGAD00001002511, EGAD00001002523, EGAD00001002732, EGAD00001001279, EGAD00001001288, EGAD00001003963, EGAD00001005060, EGAD00001005335, EGAD00001006220, EGAD00001002527, EGAD00001002756, EGAD00001002750, EGAD00001002751, EGAD00001002752, EGAD00001002753, EGAD00001002754, EGAD00001002755, EGAD00001002757, EGAD00001002758, EGAD00001003468, EGAD00001003469, EGAD00001003470, EGAD00001003475, EGAD00001003478, EGAD00001009789 (described in Supplementary Data 2). All data supporting the findings described in this manuscript are available in the article and in the Supplementary Information and from the corresponding author upon request. Source Data are provided with this paper.

## Code availability

Code is available at github.com/nloyfer/wgbs_tools (https://doi.org/10.5281/zenodo.8164926), github.com/nloyfer/MarkovDeconv (https://doi.org/10.1172/jci.insight.156529) and github.com/nloyfer/UXM_deconv (https://doi.org/10.1038/s41586-022-05580-6).

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

## Acknowledgements

This works was supported in part by funding from the National Institutes of Health USA (T32-CA009686 to M.E.M. and S.S.J., F30-CA250307 to M.E.M., R01-AI132389 to A.K., R01-CA231291 to A.W. and P30-CA51008 to A.W.). The graphical abstract and sample collection schemas in this manuscript were created with BioRender.com. We also acknowledge the following consortia that generated data used in this study: *KNIH* - This study makes use of data generated by the Korea Epigenome Project. A full list of the investigators who contributed to the generation of the data is available from 152.99.75.168/KEP. Funding for the project was provided by KOREA EPIGENOME PROJECT. *AMED-CREST/NBDC* - A part of the data used for this research was originally obtained by AMED-CREST International Human Epigenome Consortium (IHEC) research project/group led by Prof./Dr. Yae Kanai and by Core Research and Evolutional Science and Technology (CREST), International Human Epigenome Consortium (IHEC) project/group led by Hiroyuki Sasaki, both available at the website of the National Bioscience Database Center (NBDC; http://biosciencedbc.jp/en/) of the Japan Science and Technology Agency (JST). *CUHK CNARG* - This study makes use of data generated by The Chinese University of Hong Kong (CUHK) Circulating Nucleic Acids Research Group, as reported by Cheng THT et al. Clin Chem. 2019 Jul;65(7):927-936). *ENCODE* - We downloaded the call sets form the ENCODE portal (https://www.encodeproject.org/) with the following identifiers: ENCSR656TQD, ENCSR328TBS, ENCSR116JEF, ENCSR116JEF, ENCSR999RWT, ENCSR577RUU, ENCSR267SNS, ENCSR899RXH, ENCSR258MDR, ENCSR641SDF, ENCSR835OJU, ENCSR442FIY, ENCSR893RHD, ENCSR784VGW, ENCSR211VXF, ENCSR749COU, ENCSR733ZTZ, ENCSR728AGC. *CEEHRC* - The results published here are in part based upon data generated by The Canadian Epigenetics, Epigenomics, Environment and Health Research Consortium (CEEHRC) initiative funded by the Canadian Institutes of Health Research (CIHR), Genome BC, and Genome Quebec. Information about CEEHRC and the participating investigators and institutions can be found at http://www.cihr-irsc.gc.ca/e/43734.html. *DEEP* - This study makes use of data generated by the Deutsches Epigenom Programm DEEP. A full list of the investigators who contributed to the generation of the data is available from the consortium website (www.deutsches-epigenom-programm.de). *Blueprint* - This study makes use of data generated by the Blueprint Consortium. A full list of the investigators who contributed to the generation of the data is available from www.blueprint-epigenome.eu. Funding for the project was provided by the European Union's Seventh Framework Programme (FP7/2007-2013) under grant agreement no 282510 – BLUEPRINT.

## Author contributions

K.O., V.M., Y.C., D.P. and A.K. facilitated collection and processing of enrolled liver transplant patient samples and corresponding clinical data. M.E.M. performed bisulfite-capture sequencing library preparation. M.E.M. and A.J.K. purified cell-types and generated reference liver cell-type-specific epigenome data. A.J.K. and M.O.S. performed RNA-sequencing analysis. M.E.M. and S.S.J. performed bioinformatics analysis of NGS data. A.P.M. advised on statistical and computational data analysis. M.E.M. and A.W. wrote the manuscript and generated all figures. M.E.M., A.W., A.T.R. and A.K. conceived the study design and provided interpretation of results. All authors critically reviewed and approved the manuscript.

## Competing interests

A.W., A.K., and M.E.M. are named as inventors on pending patent applications (U.S. Patent Application No. 18/291,113 and U.S. Patent

Application No. 63/714,126) filed by Georgetown University, which cover the detection of liver damage using the methods described in this manuscript. The remaining authors declare that the research was conducted in the absence of any commercial or financial relationships that could be construed as a potential conflict of interest.

## Ethics approval and consent to participate

All collaborators of this study have fulfilled the criteria for authorship required by Nature Portfolio journals and have been included as authors, as their participation was essential for the design and implementation of the study. Roles and responsibilities were agreed among collaborators ahead of the research. This work includes findings that are locally relevant, which have been determined in collaboration with local partners. This research was not severely restricted or prohibited in the setting of the researchers, and does not result in stigmatization, incrimination, discrimination or personal risk to participants.
