## [Transparent Peer Review file · Nature Communications]

Circulating, cell-free DNA methylation patterns indicate cellular sources of allograft injury after liver transplant

Corresponding Author: Professor Anton Wellstein

Version 0:

Reviewer comments:

Reviewer #1

(Remarks to the Author)

McNamara and colleagues present a novel approach to characterize allograft injury in liver transplantation. This life-saving treatment offer new hope to unfortunate patients with advanced liver disease. Unfortunately, liver injury can occur, limiting the benefit. Current methods of detection are either invasive or demonstrate poor specificity to identify the phenotype of allograft injury. Without this important information, physicians may delay potential therapies and worsen patients prospects of benefit. This proof-of-concept paper proposed by McNamara and colleagues may address this problem. The investigators present a cfDNA approach that that identify subtypes of liver injury, thereby, characterizing phenotypes of liver injury. Please allow that I congratulate the authors for this approach. Specific strengths of the manuscripts include:

1. Availability of serial samples within the same patients to provide trends in subcategories of liver injury overtime and by phenotype of disease.
2. Tailored library that captured subtypes of liver-derived cfDNA allowing the investigators to characterize the source of liver injury with different phenotypes of liver injury. Indeed, liver-specific cfDNA determined via this approach correlated with biochemical measures of liver injury such as AST/ALT.
3. The availability of patients with different phenotypes of liver injury: this allowed them to validate their methods.

However, there are some concerns with this manuscript:

1. The use of serum and not plasma: The preparation of serum in traditional methods often yield lysis of cells, particularly from myeloid cells leading to contamination of genomic DNA. The degree of myeloid lysis during serum preparation can be quite variable. As the investigators show in Suppl Figure 5a, the myeloid fraction in pair serum/plasma showed variance in many samples. It is worth noting that cfDNA derived from non-hematopoietic tissue types, which is the focus of this work is not expected to vary between serum and plasma. The investigators go on to report liver-derived cfDNA in matched pair plasma and serum, noting significant correlation. The use of fraction may potentially stochastically underestimate the fraction of non-hematopoietic cfDNA, the focus of this study.
 - a. In Figure 4: in addition to representing fractions, could the investigators report the liver-derived cfDNA in copies or Geq?
 - b. Discussion: The investigators should include use of serum as a limitation to this study. Suppl. Figure 5a, Day 7 pair serum/plasma highlight a potential discrepancy that can occur with use of serum.
 - c. Figure 5D and f: Is the Geq for liver cfDNA = 0. Please clarify given that in healthy patients, liver cfDNA can make up significant fraction of cfDNA... (PMID: 30498206 and PMID: 30498206).
 - d. Figure 5 B compares total cfDNA in injury versus no injury, I assume this is total cfDNA. Given the prospects of genomic DNA contamination, I would recommend the investigators move this figure to supplement or completely remove the figure. As stated above, serum is not an ideal sample to measure cfDNA-free from contamination. This dilutes the prospects of this important work. Rather, the investigators should include in their discussion the need for future studies to compare total cfDNA, mt cfDNA and cfDNA from hematopoietic cells by injury vs. no np injury and by phenotypes of liver injury.
2. Allograft injury: could the investigators classify what type of liver injury was noted in the biopsies of this patient. Was it acute cellular rejection, antibody-mediated rejection or another phenotype of non-specific allograft injury and comment on whether the tissue injury pattern vary by injury type.
3. In Figure 6 B is confusing as it seems the Geq of myeloid cells is much lower than for hepatocytes, which the pie chart above it suggest that is not the case. Could you clarify the difference between the top and bottom with respect to myeloid cfDNA.

Reviewer #2

(Remarks to the Author)

Authors use methylation profiling of cfDNA in people that received a liver transplant, to assess the tissue origins of cfDNA. Key novel findings are

- 1) comprehensive information that has been missing so far on the methylome of human cell types of interest, related to liver biology, including stellate cells, liver endothelial cells, liver-resident immune cells and cholangiocytes;
- 2) Shortly after transplant there is a surge in cfDNA from multiple organs;
- 3) Most abnormal patterns normalize a week after transplantation;
- 4) Prolonged elevation of hepatocyte and biliary cfDNA indicates upcoming allograft injury;
- 5) cfDNA tissue of origin analysis can reveal the cause of allograft damage.

The paper addresses an important open question and an unmet clinical need. Non-invasive liquid biopsies for transplant monitoring have so far focused on donor-derived SNPs found in circulation, providing sensitive and accurate information on cell death in the transplanted organ; however information gained was limited. The approach described here has the potential to distinguish between pathological processes leading to organ damage, e.g. inflammation or drug toxicity or autoimmune hepatitis, which are actionable.

Overall, I find this to be a compelling study. Despite reservations detailed below, the key finding that liver-related cfDNA levels at day 7 post-transplant is predictive of future graft injury appears solid. Even more important, though unfortunately less developed, is the attempt to predict the mechanism of graft damage from the pattern of cfDNA.

The most important and lasting contribution of this study to the field will likely be the expansion of data on cell type-specific methylomes, along with their initial but solid characterization. This by itself would justify publication at a high profile journal. I would suggest to increase the visibility of this key element of the study (e.g. mention in abstract the specific cell types profiled), and I congratulate the authors for openly sharing the raw data with the community.

Comments

Expanded atlas. The purity of the isolated cell type is a critical aspect of the work (e.g. can the community trust that stellate cell methylomes are not vascular smooth muscle, or that purchased sinusoidal endothelial cells are not heavily contaminated with fibroblasts?). The authors assessed mostly internal consistency, i.e. that ATAC-seq, K27Ac and K4me1 signals overlap with demethylated regions in their preparations. They should provide a validation versus an external source of information regarding purity. For example – does the transcriptome obtained from each isolated cell type support its presumed identity and purity? (supplemental text mentions that this analysis was done, but I could not see the data). What are the contaminants in each preparation? Do demethylated regions overlap previously shown gene enhancers of that cell type? The information can obviously be supplementary, but is essential.

Method of cfDNA isolation. The study used serum samples and a single centrifugation step as source of cfDNA. This runs contrary to the standard use of plasma and two centrifugation steps, intended or more accurate reflection of cfDNA and minimizing contaminating genomic DNA of leukocytes.

- Please explain rationale for the unusual choice to use serum rather than plasma.

- What were the concentrations of cfDNA in normal serum samples, compared with healthy plasma? A typical concentration in healthy people i.e. 3-5 ng/ml will be reassuring that there is minimal leukocyte contamination. Data in figure 4f and 5b are worrisome (pre-transplant cfDNA levels 100x higher than normal individuals, with a huge variation). Does this reflect overwhelming contamination by genomic leukocyte DNA, or are these patients acutely sick with a massive inflammatory response? Supp

- How does cfDNA composition in normal serum samples (taken from healthy individuals using the same protocol) relate to composition in normal plasma samples? This is important for assessing the possibility that a signal originates in altered blood cell composition, vs altered cfDNA composition.

Figure 5 addresses some of these issues but must be revised.

- Please provide a straight pairwise comparison of cfDNA concentration in plasma and serum.

- Supp 5b should show a pairwise comparison.

- Supp 5e is misleading because of the use of log scale, which minimizes the large difference in the inferred presence of immune-derived cfDNA in serum vs plasma.

All in all, I worry that much of the data on myeloid and lymphoid cfDNA in the current study is an artifact resulting from blood cell lysis.

Figure 4 deconvolution. Please show also the absolute concentrations of cfDNA from individual cell types (not just fraction). This may reveal if myeloid contribution is reduced, increased or just diluted immediately after Tx.

Deconvolution.

- The paper combines all myeloid cell types and all lymphoid cell types, which is a pity since the authors have a finer resolution on these lineages. I suggest to expand.

- What is the relative contribution of B and T cell cfDNA in the acute cellular rejection?

- There is no information on the methylome of Kupffer cells and its representation in cfDNA. As this is such an important cell type, I would suggest to elaborate on findings. At a minimum, is there evidence that they truly isolated pure Kupffer cells? Which cell types do these methylomes resemble the most?

- Can the authors distinguish CNS from PNS as a source of neuronal cfDNA?
- Is cardiomyocyte cfDNA post-tx correlated with troponin? Can it be validated using targeted analysis? This is potentially a very important finding.

Drug effect. Can the authors assess the effect of immune suppression that was presumably given to recipients? Are the pre-transplant samples taken after or before treatment has started? Can this explain the abnormally high concentrations of cfDNA pre-transplant?

Minor

Title and throughout text – what is measured is not methylated DNA but methylation patterns / signatures / markers.

Is liver transplant indeed the second most common type? What about HCT?

Line 162: We were surprised to find that liver cell-type-specific hypomethylated blocks were also regions with cell-type specific chromatin accessibility and H3K27ac binding. Why were you surprised? I would change to “as expected...”.

Figure 4d-i – I believe colored lines are mean fraction, not mean % as written.

Reviewer #3

(Remarks to the Author)

Donor-derived cell free DNA hold promise to detect injury in liver transplant recipients but has not currently been associated with specific archetype of injuries. Therefore, the concept of identifying liver injury using dd-cfDNA then characterizing the etiology of the ongoing pathogenic process using dd-cfDNA methylation is of great interest and clinically relevant.

However, the conclusions of this paper are limited by the single-center and retrospective design of the study, the low number of participants enrolled in this study, the insufficient patient phenotyping and reporting of pathological lesions and diagnoses. Notably, the evaluation of cfDNA methylation associations on selected cases introduces bias in the results and hampers their generalizability to the whole population of the liver transplant recipients. Moreover, association between cfDNA methylation patterns and a standardized lesion grading (i.e Banff) should be further reported. Also, the hypothesis researchers aimed to answer must be more clearly stated as from the beginning of the paper. Hence, due these systemic issues, the overall narrative of the paper and the answers it provides are vague and the results insufficiently robust.

I hope these comments prove helpful to the authors to refine their manuscript.

Version 1:

Reviewer comments:

Reviewer #1

(Remarks to the Author)

The authors provide a detail point by point to my comments and addressed each comment comprehensively. Importantly, the authors have included a discussion on the limitation of serum in cfDNA assessments. However, the text as written should be revised. Size selection reduces large molecular weight DNA contamination. This is true. However, there are no studies detailing that this address the issue of contamination for blood cells-derived cfDNA. As the authors are aware, 1. blood has active DNAase. . In serum the DNAase is left active and could theoretically contribute to DNA fragments similar to cfDNA- which could theoretically contribute to DNA fragments similar to cfDNA.
<https://www.sciencedirect.com/science/article/pii/S0009912015000685>.

While size selection reduces large size DNA contamination, the authors could not completely exclude other potential avenues of contamination. I would recommend that authors revise this section of the discussion to maintain focus on the important findings of work while acknowledging the reality and limitations of the sample type used.

Reviewer #2

(Remarks to the Author)

The authors have addressed most of my concerns but I remain uneasy regarding the use of serum instead of plasma. I understand that this was a necessity given the retrospective nature of the samples used, but I find the response of the authors to be unsatisfactory. Thew current version of the manuscript can mislead the community on two aspects – the legitimacy of using serum samples, and the idea that pre-transplant patients have a 100-fold increase in the concentration of cfDNA. For both things, much more convincing evidence is needed. Without evidence, this is unacceptable.
Specific points:

1. What is the fraction of the 165bp fragment from the total that is taken to library preparation, after size selection? How consistent is this fraction between samples? Variation in the contribution of larger fragments, which presumably originate in blood cells, may cause artifactual variation in the estimated tissue origins, in particular of hematopoietic cells.
2. Discussion lines 344-352: the use of serum is not a "potential" limitation (suggesting that future studies may choose to use serum). It is a true, major limitation of this study. Please acknowledge this very clearly.
3. The authors write in their response: "The total cfDNA concentration reported here is similar to that of other studies analyzing cfDNA from plasma at the same timepoints following solid organ transplant (DOI:10.26508/lsa.202302003;DOI:10.1161/CIRCULATIONAHA.121.056719; DOI:10.1164/rccm.202306-1064OC; DOI:10.1172/JCI1171729)". In fact, the cfDNA concentration measured from serum in the current paper is much, much higher than the standard seen in plasma. New Supp Figure 6h shows that even in POD30 with no injury, the concentration of cfDNA is about 500ng/ml. This is 10-50 times the correct value seen in plasma in such conditions. Authors must be very clear about this point.
 - If you argue that these high concentrations reflect tissue injury in the patients studied, pre and post-transplant – please prove using plasma samples.
 - If you argue that the community can freely exchange and compare serum and plasma cfDNA measurements – please provide better evidence.
 - Alternatively, change text to clarify that serum gives much higher concentrations than plasma, and while you think this is internally consistent and therefore legitimate for your paper, this is not the way to go in future studies.

Reviewer #3

(Remarks to the Author)

I thank the authors for addressing my comments and responding to my questions. The manuscript is now clearer to me, and the subject remains highly relevant.

In my view, the critical point of this type of investigation is to demonstrate that the candidate biomarker (in this case, cell-free DNA methylation patterns) provides better diagnostic value beyond the standard of care, especially if its implementation is complex and costly, as is the case for methylation patterns.

While the study size may not allow for conclusive evidence, I believe it is important to present arguments suggesting that methylation patterns are more specific to injury patterns than conventional standard markers (e.g., ASAT, ALAT, bilirubin, alkaline phosphatase) or more recent ones (e.g., total cfDNA).

In this context, and to address other points related to the manuscript, here are my suggestions to the authors:

- I advise authors to mention the derivation of the epigenome atlas in the abstract since it is a fundamental aspect of the investigation. In my opinion, the abstract jumps too quickly to the liver transplant part although the derivation of the cell-specific epigenome represents an impressive amount of work and is the very first step of the investigation.
- Similarly, I recommend reversing paragraphs 1 and 2 of the Results section. From what I understand, the authors first established their atlas of cell-specific epigenomes and then used it to monitor transplanted patients. Presenting the methylation patterns in transplanted patients first and then returning to the development modalities disrupts the narrative. I suggest first detailing the creation of the atlas and then explaining its clinical application. This would enhance clarity and align more closely with what appears to be the chronological reality. On first reading, this was unclear to me, and I believe such adjustments can further improve the manuscript's clarity.

Clarification of "Injury"

- The term "injury" is frequently used by the authors but its definition warrants clarification, particularly for a general audience journal whose readers may not be specialists. While it is evident that it refers to tissue injury patterns (e.g., biliary, hepatocyte inflammation), the modalities (presence of inflammatory cells, quantities, etc.) and cutoffs defining injury versus borderline lesions or the absence of injury should, in my view, be developed.

cfDNA in Figure 6

- Could total cfDNA be included in the graphs for Figure 6? This would help demonstrate the added diagnostic value of studying methylation patterns beyond total cfDNA. Specifically, showing that total cfDNA levels are comparable across the three injury patterns presented by the authors, while the cfDNA composition differs, would strongly support its added value over total cfDNA.

Selection of Injury Cases

- I maintain my position that the injury cases studied by the authors are selected. These cases are often illustrative, sometimes overly "pure" (e.g., clear rejection without associated drug toxicity or recurrence of the initial pathology), and their study, while informative, does not fully encompass the spectrum of possible injuries (e.g., less pronounced rejection, cases with concurrent drug toxicity). I encourage the authors to acknowledge this limitation in their discussion.

Figures S6H and S6A

- For me, Figures S6H and S6A are key, as they, if I understand correctly, demonstrate that total cfDNA levels do not differ based on the presence or absence of allograft injury, but the composition does. These figures should be moved to the main manuscript rather than the supplementary material. Additionally, for Figure 6H, it would be helpful to specify the concentration of the measured parameter (total cfDNA if I understand correctly). Could the authors speculate on why total cfDNA levels are not elevated despite the presence of injury, given that cfDNA has been associated with injury?

Ambiguity in phrasing

- The following sentence contains redundancies that undermines the manuscript's clarity:

"We collected additional serum samples in a subset of 20 liver transplant patients to explore cfDNA changes over time during the first month after transplant, the highest risk period for post-transplant complications (Fig. 5a). Of these patients, 11 (55%) had liver biopsies showing allograft injury within the first year."

Is this referring to the first months or the first year? Please rephrase for clarity.

Additional Data for Figure 5

Could the authors also represent AST/ALT or ALP/bilirubin levels at the measured time points (pre, post, POD7, POD30) for patients with and without allograft injury? This would, in my opinion, provide a strong argument for the clinical use of cfDNA methylation patterns if the authors can demonstrate that standard routine markers fail where cfDNA methylation pattern does not. The authors highlight that there is no correlation between alkaline phosphatase or bilirubin and biliary cfDNA. If routine markers fail to indicate biliary damage in cases where cfDNA does, this would further support its clinical value.

Annotations in Figure 6

- Using identical asterisks to denote p-value significance and biopsy timing in Figure 6 is misleading. I suggest changing the annotation style and using arrows for biopsy timing.

Version 2:

Reviewer comments:

Reviewer #1

(Remarks to the Author)

I thank the authors. The authors have addressed my concerns about the use of serum instead of adequately collected plasma.

Reviewer #2

(Remarks to the Author)

The authors have addressed my remaining concerns. I support publication. Congratulations.

Point-by-point response to the reviewer comments.

Reviewer and Editorial comments are quoted in *italicized blue font*. The **Author response** provides our comments and indicates the changes made and quotes the text with the changes as appropriate.

*Line numbers provided for quoted passages below refer to those in the final unmarked manuscript.

Reviewer A (comments italicized):

(1) McNamara and colleagues present a novel approach to characterize allograft injury in liver transplantation. This life-saving treatment offer new hope to unfortunate patients with advanced liver disease. Unfortunately, liver injury can occur, limiting the benefit. Current methods of detection are either invasive or demonstrate poor specificity to identify the phenotype of allograft injury. Without this important information, physicians may delay potential therapies and worsen patients prospects of benefit. This proof-of-concept paper proposed by McNamara and colleagues may address this problem. The investigators present a cfDNA approach that that identify subtypes of liver injury, thereby, characterizing phenotypes of liver injury. Please allow that I congratulate the authors for this approach.

Specific strengths of the manuscripts include:

- 1. Availability of serial samples within the same patients to provide trends in subcategories of liver injury overtime and by phenotype of disease.*
- 2. Tailored library that captured subtypes of liver-derived cfDNA allowing the investigators to characterize the source of liver injury with different phenotypes of liver injury. Indeed, liver-specific cfDNA determined via this approach correlated with biochemical measures of liver injury such as AST/ALT.*
- 3. The availability of patients with different phenotypes of liver injury: this allowed them to validate their methods.*

Author response: We thank the reviewer for these positive comments. We are also grateful for the additional questions, which helped us to further clarify some aspects of the work, as outlined below.

(2) The use of serum and not plasma: The preparation of serum in traditional methods often yield lysis of cells, particularly from myeloid cells leading to contamination of genomic DNA. The degree of myeloid lysis during serum preparation can be quite variable. As the investigators show in Suppl Figure 5a, the myeloid fraction in pair serum/plasma showed variance in many samples. It is worth noting that cfDNA derived from non-hematopoietic tissue types, which is the focus of this work is not expected to vary between serum and plasma. The investigators go on to report liver-derived cfDNA in matched pair plasma and serum, noting significant correlation.

Author response:

We agree that the main source of HMW-DNA in serum is thought to be from cell lysis during the time the blood is allowed to clot. To account for this, we performed a size selection to enrich for fragmented cfDNA <800 bp and effectively remove HMW-DNA contamination from lysed cells (see example below). This method was previously published in Maggi et al 2018 (DOI:10.3389/fgene.2018.00006) and also in our previous publication McNamara et al 2023

(DOI:10.1172/jci.insight.156529). We copy the figure that demonstrates size selection from this publication below. The size selection helps to control for the variability in myeloid lysis during serum preparation. The same bead-based size selection was applied to all samples that were acquired through standardized serum isolation and cfDNA extraction protocols. This DNA purification step has been demonstrated to remove contaminating traces of high-molecular weight genomic DNA (~ 10 kb) from cell-free DNA released into the bloodstream through a natural process of cell death (~ 150–200 bp). In addition, we emphasize the changing solid tissue cfDNA contributions in this paper that is not expected to vary between serum and plasma. We describe this and provide the details in the Methods / Supplemental Methods and provide additional data in **Supplemental Figs. 8, 9, and 10**:

Methods (Lines 431-433): “Additional size selection using Beckman Coulter beads was applied to remove high-molecular weight DNA reflective of cell-lysis and leukocyte contamination as previously described (38, 79).”

Supplemental Materials and Methods, second para: “While plasma is produced when whole blood is collected in tubes that are treated with anticoagulant, serum is obtained after allowing blood to clot for 30 minutes at room temperature and then centrifuging the samples to remove the cellular component (2, 3). Cellular components significantly increase in serum samples that sit longer than 60 minutes; however, adherence to standard operating procedures for preparation of serum and plasma have been found to greatly reduce such contamination and sources of error (4). We took extra steps to address these concerns by ensuring timely processing of blood samples and performing an additional bead purification after cfDNA isolation to remove high-molecular weight DNA derived from blood cell lysis.”

*Figure published in McNamara et al. 2023 (DOI:10.1172/jci.insight.156529) as Supplemental Figure 13. (A, B) Representative bioanalyzer trace of freshly isolated cfDNA extracted from healthy control human serum before (A) and after (B) removal of high-molecular weight (HMW) DNA.

(2a) In Figure 4: in addition to representing fractions, could the investigators report the liver-derived cfDNA in copies or Geq? The use of fraction may potentially stochastically underestimate the fraction of non-hematopoietic cfDNA, the focus of this study.

Author response:

We report the fractions as Geq in **Supplemental Fig. 5** and also include the raw data in **Supplemental Table 8**.

The size selection to remove HMW-DNA controls for dilution of non-hematopoietic cfDNA. This is demonstrated by the striking change in fraction of cfDNA derived from liver cell-types after transplant that is homogenous across the cohort and speaks to the quality of the serum cfDNA sample preparation.

(2b) Suppl. Figure 5a, Day 7 pair serum/plasma highlight a potential discrepancy that can occur with use of serum.

Author response:

We include an additional figure to show the comparison of methylation status at the block-level in cfDNA isolated from paired human serum and plasma on POD7 only. We find that there is a high degree of correlation between the methylation status at the block-level comparing paired serum/plasma, including on POD7 (**Supplemental Fig. 9**). The variation in the estimated cell-type proportion is reduced in part by portraying the absolute liver-derived cfDNA in Geq (**Supplemental Table 8**). Also, comparing the change over time, the presence or absence of liver-derived cfDNA across the time series within the same individual is reproducible when comparing the serum/plasma pairs. We thought it supportive that the sustained elevation of hepatocyte cfDNA on POD7 was observed in all of the serum/plasma pairs of these liver transplant patients (all later diagnosed with allograft injury from tissue biopsy) (**Supplemental Fig. 8**).

Supplemental Fig. 9b, Density heatmap comparing methylation status across blocks in cfDNA isolated from paired human serum and plasma on POD7 only. Methylation levels are highly correlated at the block level with average Spearman's rho =0.763, $p < 0.05$.

(2c) Discussion: The investigators should include use of serum as a limitation to this study.

(2d) Figure 5 B compares total cfDNA in injury versus no injury, I assume this is total cfDNA. Given the prospects of genomic DNA contamination, I would recommend the investigators move this figure to supplement or completely remove the figure. As stated above, serum is not an ideal sample to measure cfDNA-free from contamination. This dilutes the prospects of this important work. Rather, the investigators should include in their discussion the need for future studies to compare total cfDNA, mt cfDNA and cfDNA from hematopoietic cells by injury vs. no injury and by phenotypes of liver injury.

Author response:

We discuss potential limitations from use of serum as the source of cfDNA and need for additional studies to further explore concentrations. We moved **Fig. 5B** to the Supplement (now **Supplemental Fig. 6h**). This figure graphs total cfDNA and we agree that further exploration of concentration and breakdown into components (i.e. mt-cfDNA) is an important future direction.

Discussion Lines (344-352): “As a potential limitation, the cfDNA used in this study was extracted from serial serum samples. To address this limitation, cfDNA was size-selected to remove contaminating HMW-DNA from cellular lysis during the process of serum preparation to allow for generalizability to other analyses of banked serum samples. The analysis of paired plasma and serum samples described in Supplemental Methods shows close correlation for the solid organ-derived cfDNAs from either source (**Supplemental Figs. 8 to 10**). However, additional studies are needed to benchmark these results against cfDNA derived from plasma collected with and without WBC-stabilizing tubes to further generalizability for clinical application.”

In this study, we adhered to a standardized protocol for serum isolation across all samples and size-selection was performed to remove any contaminating HMW-DNA. Therefore, comparison of concentration changes over time would be appropriate to reveal biological insights within and amongst this cohort. The total cfDNA concentration reported here is similar to that of other studies analyzing cfDNA from plasma at the same timepoints following solid organ transplant (DOI:10.26508/lsa.202302003; DOI:10.1161/CIRCULATIONAHA.121.056719; DOI:10.1164/rccm.202306-1064OC; DOI:10.1172/JCI171729).

(3) Figure 5D and f: Is the Geq for liver cfDNA = 0. Please clarify given that in healthy patients, liver cfDNA can make up significant fraction of cfDNA... (PMID: 30498206 and PMID: 30498206).

Author response:

We also find that hepatocyte-derived fragments are a significant contributor to the composition of cfDNA from healthy individuals, provided in **Supplemental Fig. 10g** (data reanalyzed from DOI:10.1172/jci.insight.156529, DOI:10.1038/s41586-022-05580-6, DOI: 10.1186/s13059-015-0645-x). However, we find that biliary-epithelial cfDNA is below detection in healthy individuals. In contrast to healthy individuals, hepatocyte cfDNA using this same approach was below detection in several patients after liver-transplant as early as POD7 (**Fig. 5D**). Although, it is important to note that the total liver cfDNA is not below detection in most patients after liver-transplant, encompassed by all cell-types comprising the transplanted liver tissue (**Supplemental Fig. 6e** and below). This matches what has been demonstrated looking at dd-cfDNA based on genetic differences between donor and recipient where dd-cfDNA levels stabilize after initial injury from transplant by around POD14 in heart, lung, and kidney transplants. We also find this trajectory of hepatocyte cfDNA after liver transplant to be interesting though. Possibly, this could

reflect upon aspects of remodeling and repair coupled with the effects of immunosuppressive therapy regimens that are still underexplored.

Supplemental Fig. 10g. Predicted %hepatocyte versus %biliary-epithelial derived cfDNA extracted from healthy controls (n=40). Data was obtained and reanalyzed from GSE200187, GSE186458, and phs000846

Supplemental Fig. 6e. Average of total liver cfDNA (hepatocyte, biliary-epithelial, liver-endothelial and hepatic stellate) on POD7 and POD30 (Mann-Whitney test, ns p>0.05). Serum samples from 20 liver transplant patients collected pre-transplant, post-reperfusion (POD0), post-operative day 7 and 30 (POD7, POD30). By 6 months post-transplant 9 patients showed graft acceptance and 11 patients graft injury.

(4) Allograft injury: could the investigators classify what type of liver injury was noted in the biopsies of this patient. Was it acute cellular rejection, antibody-mediated rejection or another phenotype of non-specific allograft injury and comment on whether the tissue injury pattern vary by injury type.

Author response:

The etiology of allograft injury was diagnosed from histopathological analysis of paired liver biopsy tissues (FC-bx). The data in **Fig. 5** shows the trajectory of liver cell-type damages in patients with longitudinal samples collected within the first month after transplant (prospective collection). Of this cohort 55% were diagnosed with allograft injury within the first year after transplant. We add the breakdown of injury patterns in this cohort in **Supplemental Table 6A** (column E) and also represented now in **Fig. 5B**. There wasn't sufficient sample size to compare differences in cfDNA composition grouped by etiology of allograft injury in this cohort, thus we added to this sample size in **Fig. 6** to directly compare biliary versus hepatocellular etiologies of allograft injury and cellular makeup of cfDNA.

The data in **Fig. 6** shows the cellular composition of cell-free DNA in serum samples at the same time of for cause biopsy (FC-bx) proven allograft injury in tissues. These samples were categorized into having hepatocellular or biliary etiologies of allograft injury based on the histopathological analysis of paired tissues and corresponding clinic data detailed in **Supplemental Table 6B** (column E and column F). We found a significant increase in biliary-epithelial cfDNA in samples classified by having biliary injury patterns observed in biopsies (**Figs. 6B-6D**).

(5) In Figure 6 B is confusing as it seems the Geq of myeloid cells is much lower than for hepatocytes, which the pie chart above it suggest that is not the case. Could you clarify the difference between the top and bottom with respect to myeloid cfDNA.

Author response:

We revised **Fig. 6B** and legend to depict only the solid-organ cell-type contributors and now explain the graph in more detail. The top pie chart depicts the cfDNA Geq of each cell-type averaged across all samples within each grouped etiology of allograft injury. The bottom is a contingency stacked bar graph that depicts the proportion of cellular Geq within each individual sample where each stack represents a different sample and the height of each segment within the stack representing the relative proportion of cfDNA within that cell-type group across samples. The raw cell-type proportions and Geq for each sample are also provided in **Supplemental Tables 6B and 8B**.

This graph depicts that the proportion of biliary cfDNA is higher within samples categorized as having biliary etiologies of allograft injury relative to those categorized as having hepatocellular etiologies of allograft injury.

Reviewer B (comments italicized):

(1) Authors use methylation profiling of cfDNA in people that received a liver transplant, to assess the tissue origins of cfDNA. Key novel findings are

- 1) comprehensive information that has been missing so far on the methylome of human cell types of interest, related to liver biology, including stellate cells, liver endothelial cells, liver-resident immune cells and cholangiocytes;*
- 2) Shortly after transplant there is a surge in cfDNA from multiple organs;*
- 3) Most abnormal patterns normalize a week after transplantation;*
- 4) Prolonged elevation of hepatocyte and biliary cfDNA indicates upcoming allograft injury;*
- 5) cfDNA tissue of origin analysis can reveal the cause of allograft damage.*

The paper addresses an important open question and an unmet clinical need. Non-invasive liquid biopsies for transplant monitoring have so far focused on donor-derived SNPs found in circulation, providing sensitive and accurate information on cell death in the transplanted organ; however information gained was limited. The approach described here has the potential to distinguish between pathological processes leading to organ damage, e.g. inflammation or drug toxicity or autoimmune hepatitis, which are actionable.

Overall, I find this to be a compelling study. Despite reservations detailed below, the key finding that liver-related cfDNA levels at day 7 post-transplant is predictive of future graft injury appears solid. Even more important, though unfortunately less developed, is the attempt to predict the mechanism of graft damage from the pattern of cfDNA.

The most important and lasting contribution of this study to the field will likely be the expansion of data on cell type-specific methylomes, along with their initial but solid characterization. This by itself would justify publication at a high profile journal. I would suggest to increase the visibility of this key element of the study (e.g. mention in abstract the specific cell types profiled), and I congratulate the authors for openly sharing the raw data with the community.

Author response: We thank the reviewer for the positive feedback. We appreciate the comments that have prompted additional analyses and led to improved revisions to our manuscript, as outlined below.

(2) Expanded atlas. The purity of the isolated cell type is a critical aspect of the work (e.g. can the community trust that stellate cell methylomes are not vascular smooth muscle, or that purchased sinusoidal endothelial cells are not heavily contaminated with fibroblasts?). The authors assessed mostly internal consistency, i.e. that ATAC-seq, K27Ac and K4me1 signals overlap with demethylated regions in their preparations. They should provide a validation versus an external source of information regarding purity. For example – does the transcriptome obtained from each isolated cell type support its presumed identity and purity? (supplemental text mentions that this analysis was done, but I could not see the data). What are the contaminants in each preparation? Do demethylated regions overlap previously shown gene enhancers of that cell type? The information can obviously be supplementary, but is essential.

Author response:

We include an additional figure to estimate the purity of starting cell-populations used for DNA methylation analysis based on enriched expression of genes relative to bulk liver tissue (**Supplemental Fig. 11**). We also add additional description of purity assessment in the Supplemental Methods. The cell-populations demonstrate enriched expression (over 200-fold) of lineage-specific genes relative to bulk liver tissue. The stellate cell populations do not appear to be contaminated with smooth muscle cells, based on over 16-fold lower expression of ACTA2 (smooth muscle alpha-2 actin) relative to smooth muscle cells. Quality of liver endothelial cell methylomes and specificity of identified methylation markers was also validated in a previous publication (DOI:10.1172/jci.insight.156529). Previously, the liver endothelial cells demonstrated cell-specific expression and methylation programs relative to fibroblasts in addition to other tissue-specific endothelial cell populations (liver sinusoidal versus cardio-pulmonary endothelial). Validation of biliary epithelial cells was done using DNA methylation at previously published regions with specificity to biliary tissues. This was due to poor quality of RNA extracted from these cells that prohibited RNA-sequencing analysis due to the high levels of digestive enzymes. The biliary epithelial cell-populations demonstrate purity (over 400-fold enrichment) relative to bulk liver and bulk gallbladder tissues. The majority of identified cell-type-specific hypomethylated

DMBs were characterized as enhancers, characteristic of the H3K4me1 histone modification, by chromHMM annotations in the same cell-type, providing external validation (**Fig. 3C**). Pathway analysis revealed that the cell-type-specific DMBs identified were annotated to genes relevant for cell function and identity (**Supplemental Fig. 1 and Supplemental Fig. 3**).

(3) Method of cfDNA isolation. The study used serum samples and a single centrifugation step as source of cfDNA. This runs contrary to the standard use of plasma and two centrifugation steps, intended or more accurate reflection of cfDNA and minimizing contaminating genomic DNA of leukocytes.

- Please explain rationale for the unusual choice to use serum rather than plasma.

Author response:

Prospective collection of serum samples after liver transplant for this study was started several years ago at our hospital, allowing for an expansive biobank to be established including a diversity of etiologies of allograft injury. We wanted to utilize these extensively characterized samples with corresponding clinical information. Therefore, we used a method of cfDNA extraction from serum samples that accounts for minimizing contaminating genomic DNA of leukocytes through size selection. The main source of HMW-DNA in serum is thought to be from cell lysis during the time the blood is allowed to clot. To account for this, we performed a size selection to enrich for fragmented cfDNA <800 bp and effectively remove HMW-DNA contamination from lysed cells. This method was previously published in Maggi et al 2018 (DOI:10.3389/fgene.2018.00006) and also in our previous publication McNamara et al 2023 (DOI:10.1172/jci.insight.156529) We copy the figure to demonstrate size selection from this publication below. The same bead-based size selection was applied to all samples that were acquired through standardized serum isolation and cfDNA extraction protocols. This DNA purification step has been demonstrated to remove contaminating traces of high-molecular weight genomic DNA (~ 10 kb) from cell-free DNA released into the bloodstream through a natural process of cell death (~ 150–200 bp). The size selection helps to control for the variability in myeloid lysis during serum preparation. In addition, we emphasize the changing solid tissue cfDNA contributions in this paper that is not expected to vary between serum and plasma. We describe this and provide the details in the Methods / Supplemental Methods and provide additional data in **Supplemental Figs. 8, 9, and 10**:

Methods (Lines 431-433): “Additional size selection using Beckman Coulter beads was applied to remove high-molecular weight DNA reflective of cell-lysis and leukocyte contamination as previously described (38, 79).”

Supplemental Materials and Methods, second para: “While plasma is produced when whole blood is collected in tubes that are treated with anticoagulant, serum is obtained after allowing blood to clot for 30 minutes at room temperature and then centrifuging the samples to remove the cellular component(2, 3). Cellular components significantly increase in serum samples that sit longer than 60 minutes; however, adherence to standard operating procedures for preparation of serum and plasma have been found to greatly reduce such contamination and sources of error (4). We took extra steps to address these concerns by ensuring timely processing of blood samples and performing an additional bead purification after cfDNA isolation to remove high-molecular weight DNA derived from blood cell lysis.”

*Figure published in McNamara et al. 2023 (DOI:10.1172/jci.insight.156529) as Supplemental Figure 13. (A, B) Representative bioanalyzer trace of freshly isolated cfDNA extracted from healthy control human serum before (A) and after (B) removal of high-molecular weight (HMW) DNA.

(4) What were the concentrations of cfDNA in normal serum samples, compared with healthy plasma? A typical concentration in healthy people i.e. 3-5 ng/ml will be reassuring that there is minimal leukocyte contamination. Data in figure 4f and 5b are worrisome (pre-transplant cfDNA levels 100x higher than normal individuals, with a huge variation). Does this reflect overwhelming contamination by genomic leukocyte DNA, or are these patients acutely sick with a massive inflammatory response?

Author response:

In a previous analysis of cfDNA from paired serum/plasma of non-diseased individuals (n=4), we found an average concentration of 32 ng/mL in serum and 22 ng/mL in plasma respectively (DOI:10.1172/jci.insight.156529). These concentrations reflect minimal contamination from leukocyte cell lysis after size-selection. There are also many variables that impact the concentration of cfDNA including age, exercise and inflammation (DOI:10.1016/j.xcrm.2023.101074; DOI:10.7554/eLife. 89321; DOI:10.1515/labmed-2022-0027). Recent work indicates that liver function is one of the main natural mechanisms impacting cfDNA clearance, specifically through liver-resident macrophages (DOI:10.1126/science.adf2341). Kidney function has also been found to impact cfDNA clearance. The pre-transplant cfDNA levels are 100x higher than what we measured in normal individuals. However, these individuals all have end-stage liver disease and many kidney failure as a co-morbidity. Inflammation and activation of coagulation in liver failure has been found to increase cfDNA concentration, but also the increased concentration could result from impaired clearance mechanisms. Other studies have found elevated cfDNA concentrations in individuals with impaired liver tissue function as well as demonstrated similar concentrations after solid organ transplant (DOI:10.1002/lt.25695; DOI:10.1101/2024.01.26.577500; DOI:10.1002/adv.202206789; DOI:10.1016/j.jhepr.2019.06.002; DOI:10.26508/lsa.202302003; DOI:10.1161/CIRCULATIONAHA.121.056719; DOI:10.1164/rccm.202306-1064OC; DOI:10.1172/JCI171729). We add this to the Discussion in Lines 324-331.

(5) How does cfDNA composition in normal serum samples (taken from healthy individuals using the same protocol) relate to composition in normal plasma samples? This is important for assessing the possibility that a signal originates in altered blood cell composition, vs altered cfDNA composition.

Author response:

We compared the methylation status and cellular origins of cfDNA isolated from paired serum/plasma samples of healthy human controls in a previous study, using the same protocol of extraction and size selection (DOI:10.1172/jci.insight.156529). Control human serum and plasma from healthy adult donors was purchased from Innovative Research (SKU#ISERS10ML and SKU#IPLASK2E10ML) to compare results from our analyses across sample preparations. We found that taking this approach, cfDNA methylation status at the block level is highly correlated when comparing cfDNA derived from serum or plasma (average Spearman's rho =0.70). We reanalyzed this data using the same approach as was used with the liver transplant samples in this paper and provide the data as an additional figure (**Supplemental Fig. 10**). Deconvolution analysis verified that the %immune and %solid organ origins of cfDNA does not vary across the two sample types. Thus, despite an overall higher Geq Immune found in serum due to the overall higher cfDNA concentrations, this background signal is consistent from sample-to-sample allowing for accurate comparison of changes over time in serial samples collected from the same individuals.

Supplemental Figure 10. Pairwise comparison of methylation status and cellular origins of cfDNA isolated from serum and plasma of healthy human controls. a-d, Density heatmap comparing methylation status across blocks in cfDNA isolated from paired human serum and plasma samples from healthy controls (n=4). Methylation levels are highly correlated at the block-level with an average Spearman's rho =0.70 and $R^2=0.89$. e, Immune and solid organ Geq from cfDNA isolated from serum versus plasma. f, Predicted %immune versus %solid-organ derived

cfDNA extracted from either serum or plasma. (a-f) Methylation data from paired serum and plasma samples from healthy controls reanalyzed from GSE200187. (e-f) Data presented as mean \pm SD; n=4 samples per group. Individual serum-plasma pairs are represented by differently colored dots. Wilcoxon matched-pairs signed rank test was used for comparisons amongst groups. NS, $p \geq 0.05$; * $p < 0.05$.

(6) Figure 5 addresses some of these issues but must be revised.

- Please provide a straight pairwise comparison of cfDNA concentration in plasma and serum.

- Supp 5b should show a pairwise comparison.

- Supp 5e is misleading because of the use of log scale, which minimizes the large difference in the inferred presence of immune-derived cfDNA in serum vs plasma.

All in all, I worry that much of the data on myeloid and lymphoid cfDNA in the current study is an artifact resulting from blood cell lysis.

Author response:

We revised Supplemental Fig. 8 and 9 (Previously Supplemental Fig. 5) to include these modifications. The pairwise comparison of DNA methylation at the block-level in these paired plasma and serum samples is provided below (Supplemental Fig. 9a). The individual paired serum/plasma samples are correlated when comparing DNA methylation status at the block-level across all timepoints (Supplemental Fig. 9c). There is an elevation in concentration in the serum cfDNA samples that is consistent across all samples and proportional in comparison to paired plasma (Supplemental Fig. 9d). The comparison of immune and solid-organ derived cfDNA in serum vs plasma indicates that the samples are highly correlated (Supplemental Fig. 8d and 8e).

Supplemental Figure 9a. Extended pairwise comparison of methylation status and concentration of cfDNA isolated from serum and plasma of liver transplant patients. a, Density heatmap comparing methylation status across blocks in cfDNA isolated from paired human serum and plasma (n=3 individuals each with paired serum and plasma at POD0, POD7, and POD30). Methylation levels are highly correlated at the block level with average Spearman's rho =0.795, $p < 0.05$.

Supplemental Figure 9c. Correlation of methylation status across blocks in cfDNA isolated from paired human serum and plasma (pairwise comparison). Methylation levels are highly correlated at the block level at all timepoints.

Supplemental Figure 9d. (Left) Comparison of concentration of cfDNA extracted from paired serum and plasma samples. **Supplemental Figure 8e.** (Right) Pairwise comparison of solid-organ and immune-derived cfDNA Geq in paired serum and plasma samples (colored dots represented pair-wise comparisons). While there is a slight elevation in concentration in the serum cfDNA samples relative to plasma, this elevation is consistent across all samples and proportional in breakdown of immune and solid-organ derived fragments.

(7) Figure 4 deconvolution. Please show also the absolute concentrations of cfDNA from individual cell types (not just fraction). This may reveal if myeloid contribution is reduced, increased or just diluted immediately after Tx.

Author response:

We report the fractions as Geq in **Supplemental Fig. 5** and also include the raw data in **Supplemental Table 8**.

(8) The paper combines all myeloid cell types and all lymphoid cell types, which is a pity since the authors have a finer resolution on these lineages. I suggest to expand.

- What is the relative contribution of B and T cell cfDNA in the acute cellular rejection?

Author response:

We expand the deconvolution analysis performed to estimate immune cell subsets beyond that of myeloid and lymphoid super-groups to include, mature B-cells, naïve B-cells, CD4 T-cells, CD8 T-cells, NK cells, Monocytes/Macrophages, and Neutrophils. We provide this additional data in **Supplemental Fig. 7** and **Supplemental Table 9**. After transplant there was a relative decrease in mono/macro, neutrophil, CD8 T-cell and naïve B-cell cfDNA. However, there was a significant increase in the total abundance of immune Geq from all immune cell subsets immediately after transplant comparing serial samples pre- and post-reperfusion on POD0 (**Supplemental Fig. 5**). This reflects leukocytosis from initiation of immunosuppression during the surgery, as well as inflammation related to ischemia reperfusion injury. There was not a significant difference in the immune cell composition of cfDNA in the first month after transplant comparing patients with and without allograft injury, diagnosed within the first-year post-transplant. However, assessment of changes to immune-derived cfDNA composition over time after liver transplant amongst the entire cohort, revealed slightly increased proportion of T-cell-derived cfDNA and decreased proportion of B-cell-derived cfDNA post-transplant (**Supplemental Fig. 7a**). In addition, at the time of diagnosis of allograft injury from liver biopsy there was a significantly higher composition of myeloid-derived cfDNA (both mono/macro and neutrophil) in patients with biliary etiologies of allograft injury relative to those with hepatocellular etiologies (**Supplemental Figure 7c and 7d**). We provide additional text commentary on immune cell cfDNA dynamics in the Results (Lines 246-249; 264-267), Discussion (Lines 377-382) and Supplemental Methods.

(9) There is no information on the methylome of Kupffer cells and its representation in cfDNA. As this is such an important cell type, I would suggest to elaborate on findings. At a minimum, is there evidence that they truly isolated pure Kupffer cells? Which cell types do these methylomes resemble the most?

Author response:

We include an additional figure to characterize the methylomes of the liver-resident immune cell population in more detail (**Supplemental Fig. 3**). These liver-resident immune cells were isolated from single donor healthy human liver tissues. They were purchased as cryopreserved passage 0 cells from Novabiosis where purity was analyzed using FACS analysis for %positive cell surface markers of CD14 (innate immune cell marker), CD11b (leukocyte marker), and CD68 (macrophage marker). We generated paired RNA and DNA-methylation sequencing data from these cells and validate identity as liver-resident immune cells through greater than 40-fold enrichment of immune cell gene expression relative to bulk liver tissue (**Supplemental Fig. 3C**). In addition to enriched expression of immune cell gene expression programs (TBX21, TAGAP, RUNX3, CD69, etc), we also observed enrichment of several liver-resident lymphocyte and MAIT-T-cell genes (GNLY, KLRB1, NKG7, etc) (DOI:10.1038/s41467-018-06318-7). The RNA and DNA

methylation sequencing data confirm identify as liver-resident immune cells (**Fig. 3** and **Supplemental Fig. 3**). However, they likely represent a diversity of liver-resident immune cells, rather than a purely Kupffer cell population (liver-resident macrophages). Thus, we refer to them as liver-resident immune throughout the manuscript, instead of Kupffer.

We were unable to deconvolute the origins of cfDNA with liver-resident immune cell origin. This is likely due to inter-individual variation amongst immune cell populations and lack of purity of immune cell methylomes relative to other hematopoietic cells from publicly available data. Additional large-scale efforts are needed to address this limitation. This is a future direction that we are hoping to explore more in the future. We suggest an approach combining genome-wide SNP analysis overlaid with immune cell-specific methylation patterns to establish baseline expected signal from liver-resident immune cells in cfDNA after transplant that is beyond the scope of this work.

We discuss this as a limitation in the Discussion (Lines 295-307):

“As one limitation, we were unable to identify enough DMBs with sufficient specificity to profile liver-resident immune cell turnover in patient blood samples relative to all other cell-types included in the atlas. Likely this was due to the purity and extent of peripheral and tissue-resident immune cell methylome reference data available. Instead, extended liver-resident immune cell markers were identified with relaxed specificity thresholds to use for characterization of liver cell-specific epigenetic data (**Supplemental Table 7** and Supplemental Methods). Generation of additional cell-specific methylation sequencing data to better characterize immune cell diversity will allow for enhanced ability to detect tissue-resident cell turnover in the future. In addition, implementation of hierarchical statistical models will allow for a more fine-tuned assessment of cell-free DNA composition in the face of limited numbers of highly specific methylation patterns to distinguish rare cell-types (70, 71).”

(10) Can the authors distinguish CNS from PNS as a source of neuronal cfDNA?

Author response:

This is an interesting question that we have been thinking about too. The neuron-specific methylation patterns identified in this manuscript were found using CNS neuron methylomes (**Supplemental Table 2**). However, since we don't include any PNS neurons [and this data doesn't exist for purified PNS neurons with deep-sequenced WGBS data from healthy donors], we can't be confident that this signal may be shared by other similar neuron sources not included in the atlas at present, such as enteric nervous system neurons. However, pathway analysis of the genes adjacent to identified neuron-specific methylation reinforce that these regions function in programs relevant to CNS neurons. We provide an additional supplemental figure to characterize the neuron-specific methylation patterns (**Supplemental Figure 4**). Methylation patterns were annotated to key neuron-specific genes including, NEUROG1, NODAL, NRXN2, NEURL3, GBX1, GLI2, NR2E1, OTX1, PARK2, PAX6, PLXNA4, POU3F2, and RASGRF1. Also, overlap of identified neuron-specific methylation patterns with chromatin annotations of CNS neurons from chromHMM predictions confirms that these regions are important regulatory regions in CNS neurons (see below). Thus, we decided that the most appropriate language would be neuron cfDNA and we expand on this further in the Discussion Lines 314-324.

Supplemental Figure 4b. Fraction of cell type-specific hypermethylated blocks labeled as different chromatin states in chromHMM annotations from the same cell-type (downloaded from the ENCODE project ENC539JGB).

Supplemental Figure 4a. Significant pathways related to the biological function of genes annotated to liver cell-type-specific hypomethylated blocks. Cell-type specific methylation blocks were annotated using HOMER (V4.11.1) (89) (details in the Supplemental Methods section). Pathway analysis of genes adjacent to identified tissue and cell-type specific methylation blocks was performed using Genomic Regions Enrichment of Annotations Tool (GREAT) (DOI:10.1038/nbt.1630).

(11) Is cardiomyocyte cfDNA post-tx correlated with troponin? Can it be validated using targeted analysis? This is potentially a very important finding.

Author response:

It is not part of the clinical protocol to measure troponins after liver transplant unless there is an intraoperative cardiac event. Elevated troponin levels can be misleading after liver transplant because of passive transfer from the donor tissue during the procedure (DOI:10.1016/j.bja.2018.08.024). However, other studies have demonstrated that cardiomyocyte cfDNA is correlated with troponins (DOI:10.1177/03000605241229638; DOI:10.1007/s12265-022-10295-0). We also demonstrate in a previous publication that cardiomyocyte cfDNA correlates with the maximum radiation dose to the heart in breast cancer patients treated with adjuvant radiation (DOI:10.1172/jci.insight.156529). Liver transplant is a major surgery that, during the anhepatic phase, involves clamping the supra-hepatic inferior vena cava (IVC) just inferior to the right atrium to remove the diseased liver and allow for implantation of the allograft. The clamping and the often-necessary use of vasopressor medication to maintain blood pressure can cause demand stress on the heart, which may be detected by an increase in cardiomyocyte cfDNA immediately following the procedure. Thus, we speculate that the cardiomyocyte cfDNA would correlate with an elevation in troponins after liver transplant if they had been collected.

(12) Drug effect. Can the authors assess the effect of immune suppression that was presumably given to recipients? Are the pre-transplant samples taken after or before treatment has started? Can this explain the abnormally high concentrations of cfDNA pre-transplant?

Author response:

This is an important question that we are also pursuing in further studies. Most patients are not started on immunosuppression before liver transplant, unless they have an autoimmune condition. During the transplant procedure, IV steroid bolus induction is administered for immunosuppression during the anhepatic phase, prior to the reperfusion of the allograft. This can result in leukocytosis and expansion of neutrophils. The steroids are then reduced by standard taper down to 20 mg methylprednisolone on POD6. After transplant, maintenance immunosuppression includes Tacrolimus and Mycophenolate. Complications of long-term immunosuppression after liver transplant can cause significant morbidity including renal insufficiency, hypertension, infection, malignancy, dermatologic conditions, and metabolic diseases, etc. This approach may be useful to monitor for toxicities from immunosuppression in the future and optimization of optimal dosing. In terms of sample timing, the pre-transplant samples are obtained prior to incision, and thus before the initiation of immunosuppressive treatment. In terms of high concentrations pre-transplant, please see the response to question 4 above.

Title and throughout text – what is measured is not methylated DNA but methylation patterns / signatures / markers.

Author response:

We revised the language to specify DNA methylation patterns.

Is liver transplant indeed the second most common type? What about HCT?

Author response:

We clarified that liver transplant is the second most common solid-organ transplant.

Line 162: We were surprised to find that liver cell-type-specific hypomethylated blocks were also regions with cell-type specific chromatin accessibility and H3K27ac binding. Why were you surprised? I would change to “as expected...”.

Author response:

We agree and revised the language as suggested.

It is supported by the literature that cell-type-specific hypomethylated blocks coincide with an accessible chromatin landscape and have enriched H3K27ac binding within the respective cell-type. However, we still thought that it was striking how these regions demonstrated cell-type-specificity across each of these multi-omic layers, rather than just overlapping findings across omics datasets within the same cell-type. We thought that this really demonstrates the fundamental importance of these regions to cell identity in the liver.

Figure 4d-i – I believe colored lines are mean fraction, not mean % as written.

Author response:

You are correct, we revised the legend to reflect the mean fraction represented.

Reviewer C (comments italicized):

(1) Donor-derived cell free DNA hold promise to detect injury in liver transplant recipients but has not currently been associated with specific archetype of injuries. Therefore, the concept of identifying liver injury using dd-cfDNA then characterizing the etiology of the ongoing pathogenic process using dd-cfDNA methylation is of great interest and clinically relevant.

Author response: We thank the reviewer for these positive comments. We are also grateful for the additional feedback, which helped us to further clarify some aspects of the work, as outlined below.

(2) However, the conclusions of this paper are limited by the single-center and retrospective design of the study, the low number of participants enrolled in this study, the insufficient patient phenotyping and reporting of pathological lesions and diagnoses.

Author response:

This is the largest sample size for a study of this kind to date, exploring the cellular origins of cell-free DNA fragments after liver transplant using DNA methylation patterns. First, we used 476

human WGBS samples from purified cell-types within healthy tissues to identify cell-type-specific methylation patterns. Then we translated use of the identified cell-type-specific methylation patterns to decode origins of cfDNA from serial samples collected from liver transplant patients. The longitudinal time series of samples from liver transplant patients was facilitated by a prospective (not retrospective) collection and the results are detailed in **Figs. 4 and 5**. There wasn't sufficient sample size to compare differences in cfDNA composition grouped by etiology of allograft injury in this prospective cohort, thus we added to this sample size in **Fig. 6** to directly compare biliary versus hepatocellular etiologies of allograft injury and cellular makeup of cfDNA.

The data in **Fig. 6** shows the cellular composition of cell-free DNA in serum samples at the same time of FC-bx proven allograft injury in tissues. These samples were categorized into having hepatocellular or biliary etiologies of allograft injury based on the histopathological analysis of paired tissues and corresponding clinic data detailed in **Supplemental Table 6B** (column F). Additional details on diagnoses and reporting of pathological lesions can be found in **Supplemental Table 6B** (column E and Column G-J). Additional characteristics of the liver transplant patient enrolled in this study can be found in **Supplemental Table 1**.

(3) Notably, the evaluation of cfDNA methylation associations on selected cases introduces bias in the results and hampers their generalizability to the whole population of the liver transplant recipients.

Author response:

We do not perform an association analysis of cfDNA methylation on selected cases in this study. Rather, we used 476 human WGBS samples from purified cell-types within healthy tissues to identify the cell-type-specific methylation patterns that were used as features. These 476 individuals were representative of a diverse donor population from individuals all over the world with data incorporated into the International Human Epigenome Consortium (IHEC) (**Supplemental Table 2**). Then we translated use of these pre-identified cell-type-specific methylation patterns to decode origins of cfDNA from serial samples collected from liver transplant patients. We performed extensive validation of the cell-type-specific methylation patterns used to confirm specificity and relevance to cell-type identity in **Figs 2-3** and **Supplemental Figs 1-3**. These methylation patterns are generalizable to the population at large (even beyond liver transplant) that is a major strength of this approach (DOI:10.1172/jci.insight.156529; DOI:10.1038/s41586-022-05580-6).

(4) Moreover, association between cfDNA methylation patterns and a standardized lesion grading (i.e Banff) should be further reported.

Author response:

We provide additional information about standardized lesion grading in **Supplemental Table 6B** for patients diagnosed with allograft injury from tissue biopsy. In **Figure 5**, we show that there is elevated hepatocyte and biliary epithelial cfDNA found in patients with allograft injury diagnosed within the first year after liver transplant. We provide additional data looking at lesion grading and association with liver cfDNA (Banff RAI score) in **Supplemental Fig. 6b**. There is an association between elevated RAI index with increasing liver cfDNA and decreasing immune cfDNA that is not significant, but shows a clear trend.

Supplemental Figure 6b. Association of Banff Rejection Activity Index (RAI) lesion grading of liver biopsies at time of clinical diagnosis of allograft injury with liver- and immune-derived cfDNA in the circulation

(5) Also, the hypothesis researchers aimed to answer must be more clearly stated as from the beginning of the paper. Hence, due these systemic issues, the overall narrative of the paper and the answers it provides are vague and the results insufficiently robust.

Author response:

We clarify the hypothesis and aims of this study in the Introduction and Discussion.

In the Introduction, we hypothesized that “epigenetic modifications can be used to identify cfDNA that is recipient or allograft-derived by using tissue- and cell-type specific marks that are independent of genotype differences between the donor and recipient (19–29). Allograft injury can thus be detected in an organ specific fashion, which is of critical importance.” (Lines 59-63). We strove to “utilize circulating, cell-free methylated DNA to monitor cellular damages after liver transplant, impacting the allograft tissue as well as the recipient’s organs.” (Introduction Lines 81-83).

We reiterate the overall goals of the study and major significance of the results in the Discussion (Lines 383-389): “Many studies have demonstrated the utility of donor-derived (dd) cfDNA to detect allograft injury (12, 14–17). However, dd-cfDNA is unable to discriminate amongst different causes of allograft injury. Likewise, it remains a challenge to distinguish causes relying on clinical presentation alone. Therefore, liver biopsy is still the gold standard to confirm a diagnosis and evaluate for response to treatment (4). Here we found that methylated cfDNA is able to detect and differentiate hepatocellular versus biliary causes of allograft injury at the time of biopsy-proven diagnosis (FC-bx).”

Point-by-point response to the reviewer comments.

Reviewer and Editorial comments are quoted in *italicized blue font*. The **Author response** provides our comments and indicates the changes made and quotes the text with the changes as appropriate.

*Line numbers provided for quoted passages below refer to those in the final unmarked manuscript.

Reviewer #1 (comments italicized):

(1) The authors provide a detail point by point to my comments and addressed each comment comprehensively. Importantly, the authors have included a discussion on the limitation of serum in cfDNA assessments. However, the text as written should be revised. Size selection reduces large molecular weight DNA contamination. This is true. However, there are no studies detailing that this address the issue of contamination for blood cells-derived cfDNA. As the authors are aware,

1. blood has active DNAase. In serum the DNAase is left active and could theoretically contribute to DNA fragments similar to cfDNA- which could theoretically contribute to DNA fragments similar to cfDNA.

<https://www.sciencedirect.com/science/article/pii/S0009912015000685>.

While size selection reduces large size DNA contamination, the authors could not completely exclude other potential avenues of contamination. I would recommend that authors revise this section of the discussion to maintain focus on the important findings of work while acknowledging the reality and limitations of the sample type used.

Author response: We thank the reviewer for the thoughtful response and revise the discussion as suggested.

Discussion (Lines 353-365): “As a major limitation, the cfDNA used in this study was extracted from serial serum samples. Therefore, we performed size selection of cfDNA to reduce contaminating HMW-DNA from hematopoietic cell lysis during the process of serum preparation to allow for generalizability to other analyses of banked serum samples. The analysis of paired plasma and serum samples described in Supplemental Methods shows close correlation for the solid organ-derived cfDNAs from either source (**Supplemental Figs. 8 to 10**). However, we cannot exclude other differences impacting preanalytical factors that make it challenging to compare cfDNA isolated from serum and EDTA-plasma. After size selection, we found the concentration of cfDNA in serum samples to be ~2 times elevated relative to paired plasma samples at the same timepoint. In the future, we recommend use of specialized blood collection tubes containing stabilizing preservatives to prevent cell lysis for consistent results across multi-center studies.”

Reviewer #2 (comments italicized):

The authors have addressed most of my concerns but I remain uneasy regarding the use of serum instead of plasma. I understand that this was a necessity given the retrospective nature of the samples used, but I find the response of the authors to be unsatisfactory. Their current version of the manuscript can mislead the community on two aspects – the legitimacy of using serum samples, and the idea that pre-transplant patients have a 100-fold increase in the concentration of cfDNA. For both things, much more convincing evidence is needed. Without

evidence, this is unacceptable.

Specific points:

1. What is the fraction of the 165bp fragment from the total that is taken to library preparation, after size selection? How consistent is this fraction between samples? Variation in the contribution of larger fragments, which presumably originate in blood cells, may cause artifactual variation in the estimated tissue origins, in particular of hematopoietic cells.

Author response: We provide the breakdown of fragment sizes below from analysis using a 2100 Bioanalyzer TapeStation. The fragment size profile is consistent across samples and representative of fragment size profiles of cell-free DNA from the literature (DOI:10.1172/jci.insight.144561;DOI:10.1038/s41467-024-464350; DOI:10.1126/science.aaw3616;DOI:10.1038/s41598-020-69432-x). The bead-based size selection largely reduces the contribution of larger fragments originating from lysis of hematopoietic cells during sample preparation (DOI:10.3389/fgene.2018.00006; 10.1172/jci.insight.156529).

Fragment size	Average %	SD
<100 bp	0.20	0.99
165 bp	90.46	13.61
320 bp	7.19	10.34
500-700 bp	1.90	3.87
>700 bp	0.37	1.47

We also find that the changing proportions of hematopoietic and solid tissue cell-types are relatively homogenous across the cohort comparing pre-transplant baseline to post-reperfusion changes on the day of transplant (demonstrated by data in **Fig. 4**, **Supplemental Fig. 5**, and **Supplemental Fig. 7**). This supports the quality of the samples used and effective measures taken to mitigate artifactual variation.

2. Discussion lines 344-352: the use of serum is not a “potential” limitation (suggesting that future studies may choose to use serum). It is a true, major limitation of this study. Please acknowledge this very clearly.

Author response: We revise the discussion to acknowledge the limitations of the sample type used.

Discussion (Lines 353-365): “As a major limitation, the cfDNA used in this study was extracted from serial serum samples. Therefore, we performed size selection of cfDNA to reduce contaminating HMW-DNA from hematopoietic cell lysis during the process of serum preparation to allow for generalizability to other analyses of banked serum samples. The analysis of paired plasma and serum samples described in Supplemental Methods shows close correlation for the solid organ-derived cfDNAs from either source (**Supplemental Figs. 8 to 10**). However, we cannot exclude other differences impacting preanalytical factors that make it challenging to compare cfDNA isolated from serum and EDTA-plasma. After size selection, we still found the concentration of cfDNA in serum samples to be ~2 times elevated relative to paired plasma samples at the same timepoint. In the future, we recommend use of specialized blood collection tubes containing stabilizing preservatives to prevent cell lysis for consistent results across multi-center studies.”

3. The authors write in their response: “The total cfDNA concentration reported here is similar to that of other studies analyzing cfDNA from plasma at the same timepoints following solid organ transplant (DOI:10.26508/lsa.202302003;DOI:10.1161/CIRCULATIONAHA.121.056719; DOI:10.1164/rccm.202306-1064OC; DOI:10.1172/JCI171729)”. In fact, the cfDNA concentration measured from serum in the current paper is much, much higher than the standard seen in plasma. New Supp Figure 6h shows that even in POD30 with no injury, the concentration of cfDNA is about 500ng/ml. This is 10-50 times the correct value seen in plasma in such conditions. Authors must be very clear about this point.

- If you argue that these high concentrations reflect tissue injury in the patients studied, pre and post-transplant – please prove using plasma samples.

- If you argue that the community can freely exchange and compare serum and plasma cfDNA measurements – please provide better evidence.

- Alternatively, change text to clarify that serum gives much higher concentrations than plasma, and while you think this is internally consistent and therefore legitimate for your paper, this is not the way to go in future studies.

Author response: We revise the discussion to acknowledge the limitations of the sample type used, please see the response above.

From the analyses of paired serum-plasma samples, we found the cfDNA concentration in serum samples to be ~2 times elevated relative to plasma samples across all timepoints (**Supplemental Fig. 9d**). We point to this difference in the discussion and agree that this is internally consistent within the serum samples used in this study. We encourage use of specialized blood collection tubes containing stabilizing preservatives in future studies because a multitude of parameters including timely processing of EDTA-plasma can impact sample stability and composition (DOI:10.1021/pr800545q;DOI:10.1080/15592294.2020.1827714;DOI:10.3389/fcell.2024.1385041; DOI:10.1016/j.jmoldx.2022.02.005).

Recent work indicates that liver function is one of the main natural mechanisms impacting cfDNA clearance, specifically through liver-resident macrophages (DOI:10.1126/science.adf2341). Kidney function has also been found to impact cfDNA clearance. We found the pre-transplant cfDNA levels to be ~10 times higher than what we measured in healthy individuals using serum samples and ~15 times higher than healthy individuals using plasma samples. In a previous analysis of cfDNA from paired serum/plasma of non-diseased individuals (n=4), we found an average concentration of 32 ng/mL in serum and 22 ng/mL in plasma respectively (DOI:10.1172/jci.insight.156529). It is noteworthy that liver transplant candidates suffer from end-stage liver disease, many with kidney failure as a co-morbidity. Thus, impaired clearance mechanisms of cfDNA can contribute to the elevated concentrations relative to healthy individuals.

It is not currently standard-of-care to take tissue biopsies after liver transplant without clinical indicators that warrant this, unlike routine monitoring that is done following heart transplants (DOI:10.1002/hep.22742; DOI:10.1016/j.jhep.2024.07.032; DOI:10.1056/NEJMra1610570), Thus, we define “injury” in this study to refer to patients with clinical presentation warranting a tissue biopsy and then subsequent biopsy-proven diagnosis of injury phenotype. We further clarify what we mean by the term “injury” in the Results (Lines 230-239). There is still significant remodeling and repair ongoing in all transplant recipients during this time frame, despite not having biopsy-proven tissue diagnosis of injury (DOI:10.1016/j.livres.2024.01.002; DOI:10.1007/s00268-002-4060-6). There are many variables that impact baseline cfDNA concentrations amongst individuals, making it difficult to compare static values across studies at a single timepoint. The longitudinal analyses performed here with serial samples from the same

individual over time allows for an internal control that makes comparing findings across studies more robust. We demonstrate that the trajectory of signal from various cell types over time and the changing composition of cell-free DNA associated with injury phenotypes is comparable across sample types (**Supplemental Figs. 8 and 9**).

Reviewer #3 (*comments italicized*):

I thank the authors for addressing my comments and responding to my questions. The manuscript is now clearer to me, and the subject remains highly relevant.

In my view, the critical point of this type of investigation is to demonstrate that the candidate biomarker (in this case, cell-free DNA methylation patterns) provides better diagnostic value beyond the standard of care, especially if its implementation is complex and costly, as is the case for methylation patterns.

While the study size may not allow for conclusive evidence, I believe it is important to present arguments suggesting that methylation patterns are more specific to injury patterns than conventional standard markers (e.g., AST, ALT, bilirubin, alkaline phosphatase) or more recent ones (e.g., total cfDNA).

Author response: We thank the reviewer for these positive comments. We are also grateful for the suggestions, which helped us to further clarify some aspects of the work, as outlined below.

(1) I advise authors to mention the derivation of the epigenome atlas in the abstract since it is a fundamental aspect of the investigation. In my opinion, the abstract jumps too quickly to the liver transplant part although the derivation of the cell-specific epigenome represents an impressive amount of work and is the very first step of the investigation.

Author response: We revised the abstract as suggested.

Abstract: "Post-transplant complications reduce allograft and recipient survival. Current approaches for detecting allograft injury non-invasively are limited and do not differentiate between cellular mechanisms. Here, we monitor cellular damages after liver transplants from cell-free DNA (cfDNA) fragments released from dying cells into the circulation. We analyzed 130 blood samples collected from 44 patients at different time points after transplant. **Sequence-based methylation of cfDNA fragments were mapped to an atlas of cell-type-specific DNA methylation patterns generated from 476 methylomes of purified cells.** For liver cell types DNA methylation patterns and multi-omic data integration show distinct enrichment in open chromatin and regulatory regions functionally important for the respective cell types. We find that multi-tissue cellular damages post-transplant recover in patients without allograft injury during the first post-operative week. However, sustained elevation of hepatocyte and biliary epithelial cfDNA within the first month indicates early-onset allograft injury. Further, cfDNA composition differentiates amongst causes of allograft injury indicating the potential for non-invasive monitoring and timely intervention."

(2) Similarly, I recommend reversing paragraphs 1 and 2 of the Results section. From what I understand, the authors first established their atlas of cell-specific epigenomes and then used it to monitor transplanted patients. Presenting the methylation patterns in transplanted patients first and then returning to the development modalities disrupts the narrative. I suggest first detailing the creation of the atlas and then explaining its clinical application. This would enhance clarity and align more closely with what appears to be the chronological reality. On first reading,

this was unclear to me, and I believe such adjustments can further improve the manuscript's clarity.

Author response: We revised **Fig. 1** to emphasize generation of the atlas of cell-specific DNA methylation patterns used to analyze the cell-free DNA samples collected from liver transplant patients. We tailored generation of the atlas to analyze these samples collected from liver transplant patients, generating DNA methylation, histone modification, and chromatin accessibility data from purified liver cell-types relevant to the analysis. We thought this modification in response to the reviewer's suggestion best reflects the parallel effort of atlas generation and sample collection and processing efforts leading to the results of the study.

Figure 1

(3) Clarification of "Injury"

- The term "injury" is frequently used by the authors but its definition warrants clarification, particularly for a general audience journal whose readers may not be specialists. While it is evident that it refers to tissue injury patterns (e.g., biliary, hepatocyte inflammation), the modalities (presence of inflammatory cells, quantities, etc.) and cutoffs defining injury versus borderline lesions or the absence of injury should, in my view, be developed.

Author response: We clarified the term injury and better explained cutoffs used to designate between patients with and without allograft injury.

Results (Lines 230-239): "Patients with allograft injury were diagnosed from histopathological analysis of liver biopsy tissues. Liver biopsies are not typically considered the standard of care for routine monitoring of liver transplant patients. Instead, they are usually performed only when clinically indicated.⁴ The majority of patients in this cohort designated as being "without allograft injury" did not have liver biopsies taken after transplant. Two patients were labeled "without injury" who had liver biopsies taken after transplant that were negative for pathology. These two borderline cases were both found negative for acute cellular rejection, using the standardized lesion grading, Banff Rejection Activity Index. We required an RAI index ≥ 3 to be considered ACR and designate allograft injury."

Supplemental Methods, para titled, *Classification of serum samples at time of for-cause liver biopsy (FC-bx) to diagnose allograft injury*

Serum samples from 24 liver transplant patients (n = 30 serum samples) were taken at the time of for-cause liver biopsy (FC-bx) to diagnose allograft injury. Patients were classified as having hepatocellular (n=14), biliary (n=6), or mixed hepatobiliary (n=10) etiologies of allograft injury from histopathological analysis of the liver biopsy tissues annotated by a pathologist. The following were defined as clinical etiologies of hepatocellular injury: Acute cellular rejection (ACR, rejection activity index, RAI 3+), recurrence of primary hepatic disease (including recurrence of viral hepatitis (HBV or HCV), autoimmune hepatitis, or (non)-alcoholic steatohepatitis in the transplanted organ), drug-induced hepatotoxicity, ischemia-reperfusion injury (IRI) leading to ischemic hepatitis. The following were defined as clinical etiologies of biliary injury: anastomotic and non-anastomotic biliary strictures, ascending cholangitis, ischemic cholangiopathy, recurrence of primary biliary disease (including primary sclerosing cholangitis (PSC) among others), and septic cholestasis (7–11). Mixed hepatobiliary forms of allograft injury were characterized by diagnosis of one or more hepatocellular and one or more biliary forms of allograft injury at the same timepoint.

Liver allograft injury was assessed using standardized histological criteria based on established classifications. Acute Cellular Rejection (ACR) was characterized by portal, ductal, and endothelial inflammation, with severity graded as mild, moderate, or severe based on the extent of damage (12). Antibody-Mediated Rejection (AMR) was identified using a combination of histological features, including microvascular inflammation, C4d staining in portal or sinusoidal vessels, and evidence of donor-specific antibodies (DSAs) (13). Ischemic hepatitis was diagnosed by the presence of centrilobular necrosis and minimal inflammatory infiltrates, consistent with reduced blood flow to the liver (14). Biliary complications were evaluated based on characteristic findings. Cholangitis was identified by neutrophilic infiltrates in bile ducts, while biliary strictures and bile leaks were associated with ductal narrowing, bile duct proliferation, cholestasis, and evidence of bile extravasation (15, 16). While the Banff classification primarily focuses on rejection-related injuries, these criteria provided a framework for recognizing and grading various etiologies of liver allograft injury."

Additional information about tissue injury patterns leading to histopathological diagnosis (necrosis, fibrosis, cholestasis, steatosis, RAI scores, etc.) are found in **Supplemental Table 6b**.

(4) Could total cfDNA be included in the graphs for Figure 6? This would help demonstrate the added diagnostic value of studying methylation patterns beyond total cfDNA. Specifically, showing that total cfDNA levels are comparable across the three injury patterns presented by the authors, while the cfDNA composition differs, would strongly support its added value over total cfDNA.

Author response: We added the total cfDNA concentrations corresponding to the samples in **Fig.6** to **Supplemental Fig. 7c**. We did not find a significant difference in total concentration comparing samples with hepatocellular versus biliary etiologies of allograft injury. Although, additional studies with larger sample size are needed to explore this further. Especially since the concentration of cfDNA changes over time post-transplant and the onset of injury in these different etiologies can differ, making time an important variable to control for in the future. For the samples used in this study, there was a difference in the average number of days post-transplant that hepatocellular and biliary etiologies of injury were diagnosed (see below). The higher concentration cfDNA sample outliers in the hepatocellular and mixed injury groups had injury diagnosed closer to the date of transplant surgery. The left graph is from **Suppl Fig. 7c**, the data from the right graph is from **Suppl Table 6b**.

There is a large diagnostic value in separating donor-derived cfDNA from total cfDNA due to the many variables that can impact overall cfDNA abundance (DOI:10.3389/fcell.2024.1385041; DOI:10.1016/j.xcrm.2023.101074; DOI:10.7554/eLife. 89321; DOI:10.1515/labmed-2022-0027). Both genetic SNPs and epigenetic approaches can be used to separate donor-derived cfDNA fragments. However, DNA methylation patterns that are cell-type-specific have the added value of differentiating amongst different etiologies of allograft injury based on changing cfDNA composition. We found that cfDNA fragments originating in liver epithelial cell-types had more diagnostic value as compared to total liver-derived cfDNA fragments (**Fig. 5c-g** and **Supplemental Fig. 6c,d,e**).

We emphasize this point in the Discussion (Lines 398-416):

“Many studies have demonstrated the utility of donor-derived (dd) cfDNA to detect allograft injury (12, 14–17). However, dd-cfDNA is unable to discriminate amongst different causes of allograft injury. Likewise, it remains a challenge to distinguish causes relying on clinical presentation alone. Therefore, liver biopsy is still the gold standard to confirm a diagnosis and evaluate for response to treatment (4). Here we found that cfDNA methylation is able to detect and differentiate

hepatocellular versus biliary causes of allograft injury at the time of biopsy-proven diagnosis (FC-bx). Biliary complications after liver transplant, such as ascending cholangitis, strictures (both anastomotic and non-anastomotic), leaks, and recurrence of primary sclerosing cholangitis, contribute significantly to post-transplant morbidity and mortality in both living and deceased donor transplant recipients. Conventional diagnostic and monitoring methods for these conditions often necessitate cross-sectional imaging techniques, such as MRCP, or invasive procedures like ERCP or liver biopsy, which pose additional risks to patients (77, 78). Enhanced detection of biliary cell-type-specific damage allows for differentiation from hepatocellular forms of allograft injury and associated tissue damage. This enables an earlier and more accurate diagnosis of biliary complications and improved non-invasive monitoring post-treatment. Incorporating cfDNA as a diagnostic tool into clinical practice could potentially reduce the need for invasive procedures and facilitate early intervention with targeted treatment.”

(5) Selection of Injury Cases

- I maintain my position that the injury cases studied by the authors are selected. These cases are often illustrative, sometimes overly "pure" (e.g., clear rejection without associated drug toxicity or recurrence of the initial pathology), and their study, while informative, does not fully encompass the spectrum of possible injuries (e.g., less pronounced rejection, cases with concurrent drug toxicity). I encourage the authors to acknowledge this limitation in their discussion.

Author response: We acknowledge this as a limitation in the discussion. We prospectively collected the samples for longitudinal analysis in **Fig. 5**. However, there wasn't sufficient sample size to compare differences in cfDNA composition grouped by etiology of allograft injury in this cohort, thus we added samples in **Fig. 6** to directly compare biliary versus hepatocellular etiologies of allograft injury and cellular makeup of cfDNA. These cases were selected for having tissue-biopsy proven etiologies of allograft injury and paired serum and tissues samples collected at the same time.

Discussion (Lines 402-420):

“Here we found that cfDNA methylation is able to detect and differentiate hepatocellular versus biliary causes of allograft injury at the time of biopsy-proven diagnosis (FC-bx). ...However, only samples having tissue-biopsy proven etiologies of allograft injury and paired serum and tissues samples collected at the same time were used in this study. Additional large-scale studies are needed to fully encompass the spectrum of possible injuries and also to fully elucidate the capacity of cfDNA methylation patterns to predict and diagnose injury earlier than existing biochemical markers of injury.”

(6) Figures S6H and S6A

- For me, Figures S6H and S6A are key, as they, if I understand correctly, demonstrate that total cfDNA levels do not differ based on the presence or absence of allograft injury, but the composition does. These figures should be moved to the main manuscript rather than the supplementary material. Additionally, for Figure 6H, it would be helpful to specify the concentration of the measured parameter (total cfDNA if I understand correctly). Could the authors speculate on why total cfDNA levels are not elevated despite the presence of injury, given that cfDNA has been associated with injury?

Author response: Yes, we agree. We did not find a significant difference in total cfDNA concentration comparing patients with and without allograft injury at any of the timepoints collected in the first month after transplant (**Supplemental Fig. 6h**). We edited the legend in **Supplemental Fig. 6h** to clarify that total cfDNA is the parameter being measured. However, we

did find a difference in cfDNA composition (**Supplemental Fig. 6a**). The data depicted in **Supplemental Fig. 6a** as fractions is also portrayed in **Fig. 5** as genome equivalents, a measure that takes into account the estimated cell-type proportion and the concentration of a given sample.

The abundance of total cfDNA is associated with injury, but it is also found to be elevated in response to other physiologic changes as well, such as with exercise, pregnancy and aging. There are a multitude of factors that have been demonstrated to influence cfDNA concentrations, including gender, diet, inflammation, obesity, infection, diabetes, and stress ((DOI:10.3389/fcell.2024.1385041; DOI:10.1016/j.xcrm.2023.101074; DOI:10.7554/eLife. 89321; DOI:10.1515/labmed-2022-0027). In addition, there is remodeling and repair that occurs after liver transplant that can be normal tissue regeneration and still impact cfDNA concentration in the absence of biopsy-proven etiologies of liver injury. For example, we did not find a significant difference in hepatic stellate or endothelial cfDNA levels in patients with and without allograft injury, despite contributing to the total liver-derived signal. We previously moved the data in **Supplemental Fig. 6h** to the supplement per the request of the other reviewers because we used serum samples for this study that have a higher cfDNA concentration as compared to cfDNA extracted from EDTA-plasma. We wanted to emphasize the changing abundance of specific cell-types from solid tissues over time that is more generalizable across studies as opposed to the overall abundance.

(7) Ambiguity in phrasing

- The following sentence contains redundancies that undermines the manuscript's clarity: "We collected additional serum samples in a subset of 20 liver transplant patients to explore cfDNA changes over time during the first month after transplant, the highest risk period for post-transplant complications (Fig. 5a). Of these patients, 11 (55%) had liver biopsies showing allograft injury within the first year."

Is this referring to the first months or the first year? Please rephrase for clarity.

Author response: We revise the sentence to clarify as suggested.

Results (Lines 225-230): "We collected additional serum samples in 20 liver transplant patients to explore cfDNA changes over time at defined timepoints during the first month after transplant, the highest risk period for post-transplant complications (**Fig. 5a**). Of this cohort, 11 patients (55%) had liver biopsies showing allograft injury. For 5 of the 11 patients, these biopsies were within the first month overlapping with the timing of serum sample collection. For the remaining 6 patients the biopsies were within the first year after transplant."

(8) Additional Data for Figure 5

-Could the authors also represent AST/ALT or ALP/bilirubin levels at the measured time points (pre, post, POD7, POD30) for patients with and without allograft injury? This would, in my opinion, provide a strong argument for the clinical use of cfDNA methylation patterns if the authors can demonstrate that standard routine markers fail where cfDNA methylation pattern does not. The authors highlight that there is no correlation between alkaline phosphatase or bilirubin and biliary cfDNA. If routine markers fail to indicate biliary damage in cases where cfDNA does, this would further support its clinical value.

Author response: We added this data in **Supplemental Fig. 12**. There was not a significant difference between AST/ALT or ALP/bilirubin levels comparing patients with and without allograft injury, although there was a trend towards elevated lab values in patients with allograft injury. We agree that this is a strength of cfDNA methylation patterns and have highlighted this further in the

discussion (Lines 376-377). As compared to standard biochemical markers, cfDNA has a short half-life between 15 minutes and 2 hours and one can estimate the genome equivalents of cellular turnover at the time of collection using cell-type-specific methylation patterns.

Supplemental Figure 12

Supplemental Figure 12. Trends in liver function test values over time in patients with and without allograft injury following liver transplantation. (a-f) Relationship between time and

ALT (alanine aminotransferase), AST (aspartate aminotransferase), ALP (alkaline phosphatase), Total Bilirubin, Direct Bilirubin, and INR levels measured at multiple time points post-transplant. Patients were stratified based on the presence or absence of allograft injury, determined by histopathological analysis of liver tissue biopsies within the first-year post-transplant. The solid curve represents the locally weighted regression smoothing (LOESS) trend, highlighting the mean of the fitted values. The shaded region indicates the 95% confidence interval for the LOESS fit. Colored lines indicate patients with allograft injury, and black lines represent those without allograft injury. Wilcoxon rank-sum tests were performed at each timepoint to assess differences in values between the two groups. Wilcoxon rank-sum tests were also performed to assess differences in the mean rate of change in values between the two groups. No statistically significant differences were observed at any timepoint ($p > 0.05$).

(9) Annotations in Figure 6

- Using identical asterisks to denote p-value significance and biopsy timing in Figure 6 is misleading. I suggest changing the annotation style and using arrows for biopsy timing.

Author response: We changed the annotation style in **Fig. 6** to use arrows for biopsy timing.